# The changing impact of rural electrification on Indian agriculture

Sudatta Ray [1,2] & Hemant K. Pullabhotla [3]

Rural electrification policies in the developing world primarily focus on household power, often at the cost of electricity supply to other productive sectors of the economy. We examine the consequences of this imbalance in rural electrification policy priority on agricultural development in India. Electric pumping of groundwater for irrigation is a major driver of India's agricultural growth. However, the government of India shifted its rural electrification focus towards universal household electrification starting early 2000s. Using a newly constructed panel-dataset spanning three decades, we find that districts electrifying after the policy change experience much lower gains in electrified groundwater irrigation. On average, electrifying 100 additional rural households is associated with an increase of two additional electrified wells among newly electrified districts – eight times lower compared to 16 electrified wells per 100 electrified households among districts electrified pre-policy change. Our estimates imply that newly electrified districts would have witnessed nearly 20% more irrigated cropland in the dry season if rural electrification policy priorities had not shifted away from agriculture. These results highlight the need to complement household electrification with powering income-generating sectors of the rural economy.

Electrification is deemed essential for development and household well-being. However, how electricity benefits households, particularly household income and expenditure, is less understood. This is true for India, where recent gains in rural electrification (RE) have been vast and impressive, but empirical evidence suggests limited impacts on household incomes and expenditures[1–3]. Literature on impacts of RE on household income rarely focus on the specific pathways through which households consume electricity to generate income. In rural India, where agriculture still drives the economy, the primary use of electricity for generating income is through pumping groundwater for irrigation. Roughly two-thirds of all irrigated land in India is currently based on groundwater consuming nearly 200 TWh of electricity annually (~20% of total electricity sales)[4,5]. Pumped irrigation is among the most important determinants of prosperity among smallholder farmers in India[6]. This vast consumption of groundwater has not been without its adverse impacts. Every year India pumps twice as much groundwater as the US or China, and houses regions with the greatest

rates of global groundwater depletion[7–9]. However, not all regions irrigate using groundwater, even in the presence of high availability and healthy water tables. In these regions, electric pumping of groundwater remains limited despite significant gains in RE. In this paper, we explore the reasons behind this disconnect.

We construct a novel district-level panel dataset linking household electrification, the most popular metric for measuring RE, electrification of groundwater wells, and groundwater irrigation spanning 1986 to 2013 across India. We use this dataset to understand why RE serves agriculture only in select regions across India. We find that the regional selection in electrified groundwater pumping can be attributed largely to a shift in the target of RE policy from agriculture to households which took place in India in the early 2000s. Our results imply that the new electricity infrastructure built after the policy shift constrains electric pumping of groundwater. Recent evidence suggests that gains from RE are often concentrated in specific subgroups of households[10,11]. Our results suggest the ability to use electricity for

[1]Department of Geography, National University of Singapore, Singapore, Singapore. [2]Environmental Studies, Division of Social Science, Yale-NUS College, Singapore, Singapore. [3]Department of Economics, Deakin Business School, Burwood, Australia. e-mail: sray@nus.edu.sg

irrigation as a potential explanation for these heterogeneous effects. Targeting households for RE, though necessary, potentially limits the economic impacts of electrification as most economic activities occur outside homes in India and across the developing world. Perhaps it is time to reevaluate domestic electrification as the primary target of RE policies, and instead consider income generation as a primary goal to unleash the true power of electrification in the rural developing world.

Electricity access has been gaining importance as a driver of development since before its adoption among the Sustainable Development Goals in 2015 (see SDG Goal 7.1.1). Electricity access is primarily a rural challenge due to high infrastructure costs and low payment rates. In response, government-subsidized electrification programs have mushroomed in recent years, often aided by multilateral development funds. The World Bank alone had provided more than USD 5 billion for electrification programs across 35 countries between 2010 and 2018[12]. Household electrification has been the primary target of such electricity access programs in most developing countries. In India, a similar focus on household electrification preceded the announcement of UN SDGs. In 2005, the central government launched the *Rajiv Gandhi Grameen Vidyutikaran Yojana* (RGGVY) to provide financial support to states lagging in household electrification. RGGVY has been acclaimed as one of the most extensive and successful programs in rural electrification[13]. The latest government numbers indicate near universal household electrification (see Saubhagya dashboard).

However, little can be conclusively said about the short-run or long-run impacts of RGGVY in particular and of electrification programs more generally. In India, no meaningful impact of RGGVY-led electrification was found on male or female employment or household consumption[1]. There is also scant evidence to suggest electrification leads to significant economic gains in newly electrified regions of South Africa[14], Kenya[15], and Rwanda[16]. These studies suggest that electrification may not lead to economic gains at previously assumed scales. They also highlight our lack of knowledge about the mechanisms through which electrification impacts household incomes and other economic outcomes. As a recent review of the impact of electrification in low-income countries notes, "access to household electrification alone is not enough to drive meaningful gains in development outcomes."[11] Here, we fill this gap by studying the impact of RE on economic development through agriculture, which remains the largest employment sector of the Indian rural economy.

Agriculture continues to be the mainstay of India's rural economy employing over 70% of the country's rural working population[17]. India's peculiar case of structural transformation over the last three decades has been characterized by a shift from increasing returns to agriculture to tremendous increases in services, skipping manufacturing altogether. Although this shift has been marked by spectacular GDP growth rates at the national level, it has been accompanied by disappointing outcomes in rural employment and permanent rural-urban migration[18]. Therefore at least in the near-term, increasing farm incomes remains an important avenue for addressing rural poverty. Further, pumped irrigation is among the most important determinants of prosperity among Indian smallholder farmers[6].

Irrigation in India is primarily sourced from groundwater because of its relatively low variable cost of extraction (provided the presence of electricity infrastructure) and individual accessibility. Groundwater can be accessed during dry cultivation months (the *Rabi* season) when surface water flow is diminished, making it a particularly effective source of irrigation. Additionally, wells afford farmers independence, as wells, unlike canals, do not suffer from competing uses of hydropower generation and riparian rights. Groundwater wells have been increasing in India since data on them was first collected during 1986–87. The most recent round of data indicates that growth between 2005–06 and 2013–14 occurred almost entirely in the number of deep tube-wells, indicative of increasing depths to water in some parts of

India[19]. Deep tube-wells are defined as those with depths greater than 70m and consequently require submersible pump-sets that run exclusively on electricity. Electricity is preferred even in shallower wells where the option of pumping using diesel exists and is reflected in the declining proportion of wells operated by diesel pumps from over 30% in 1986 to below 28% in 2013, despite greater than 90% growth in the total number of irrigation wells during the same period[20]. At greater than INR 40/l since price deregulation in 2014, diesel is more expensive than agricultural electricity rates in most Indian states (assuming the energy content of diesel to be 38.9 MJ/l, INR 40/l translates to approximately INR 4/kWh)[21,22]. Therefore, only in regions where electricity is unavailable or suffers from quality issues of intermittency and low voltage, diesel pumps are used to power irrigation either exclusively, or in conjunction with electric pumps to compensate for unreliable electricity supply[23,24]. Where available, electricity is subsidized by way of flat or no tariffs effectively eroding the marginal cost of consuming electricity[23]. Consequently, cheap electricity to agriculture has given rise to a strong groundwater-electricity nexus that is difficult to break out of due to political and electoral considerations, even at the cost of diminishing groundwater resources. Government procurement of rice and wheat creates strong incentives to grow water-intensive staples and further tightens the proverbial Gordian knot of this nexus. The issue is both well documented and continues to be studied for politically feasible alternatives[25-31].

Yet, there are parts of India where groundwater levels are healthy but limited groundwater irrigation constraints dry season cultivation. Many districts have a large share of replenishable groundwater available for irrigation (Fig. 1a) but a low density of electrified wells (Fig. 1b). As a consequence, these districts see a much smaller share of cultivated area under irrigation (Fig. 1c). A high water deficit during the winter dry season (Fig. 1d) adds to the low levels of winter cultivation. The variation in groundwater irrigation across India is not reflective of its abundance. Rather, the number of electrified wells may hold the key to unlocking the potential of groundwater irrigation in these regions. In this paper, we explore why electricity does not serve groundwater irrigation in some parts of India.

India is often heralded as the poster-child of RE success across the developing world[32]. Latest available government data on household electrification indicates complete coverage of all rural households (see Saubhagya dashboard). However, it was not until the late 1960s that the national government focused on RE when Green Revolution (GR) entered India (Fig. 2).

Two large-scale famines in 1965 and 1966, coupled with President Lyndon Johnson's short tether policy on PL-480, created an urgent need for India to gain self-sufficiency in grain production[33,34]. Although high-yielding variety seeds, along with chemical fertilizers and pesticides, formed the pillars of India's GR, reliable and steady sources of irrigation were key to GR's success. Areas with assured irrigation, which were primarily served by canals, were initially targeted as part of the "betting on the strong approach"[35]. Groundwater irrigation soon expanded in response to the stagnating development of canal irrigation[36]. Groundwater irrigation intensification was hinged on cheap motive power, which ultimately fueled RE's expansion in rural areas, that benefited most from GR.

Regions that enjoyed GR-led increases in agricultural production also witnessed increasingly politicized agrarian lobbies who demanded better and cheaper inputs to further aid agricultural production[35]. In response, not only did agricultural electricity coverage increase, but electricity rates were also subsidized by instituting flat tariffs in states which witnessed the greatest returns from GR technology[37]. Household electrification was secondary to agricultural electricity at this time[13]. By the early 1990s, the national government was no longer concerned about grain production, with the result that RE lagged behind in states which had lost out on GR-driven demand for electricity.

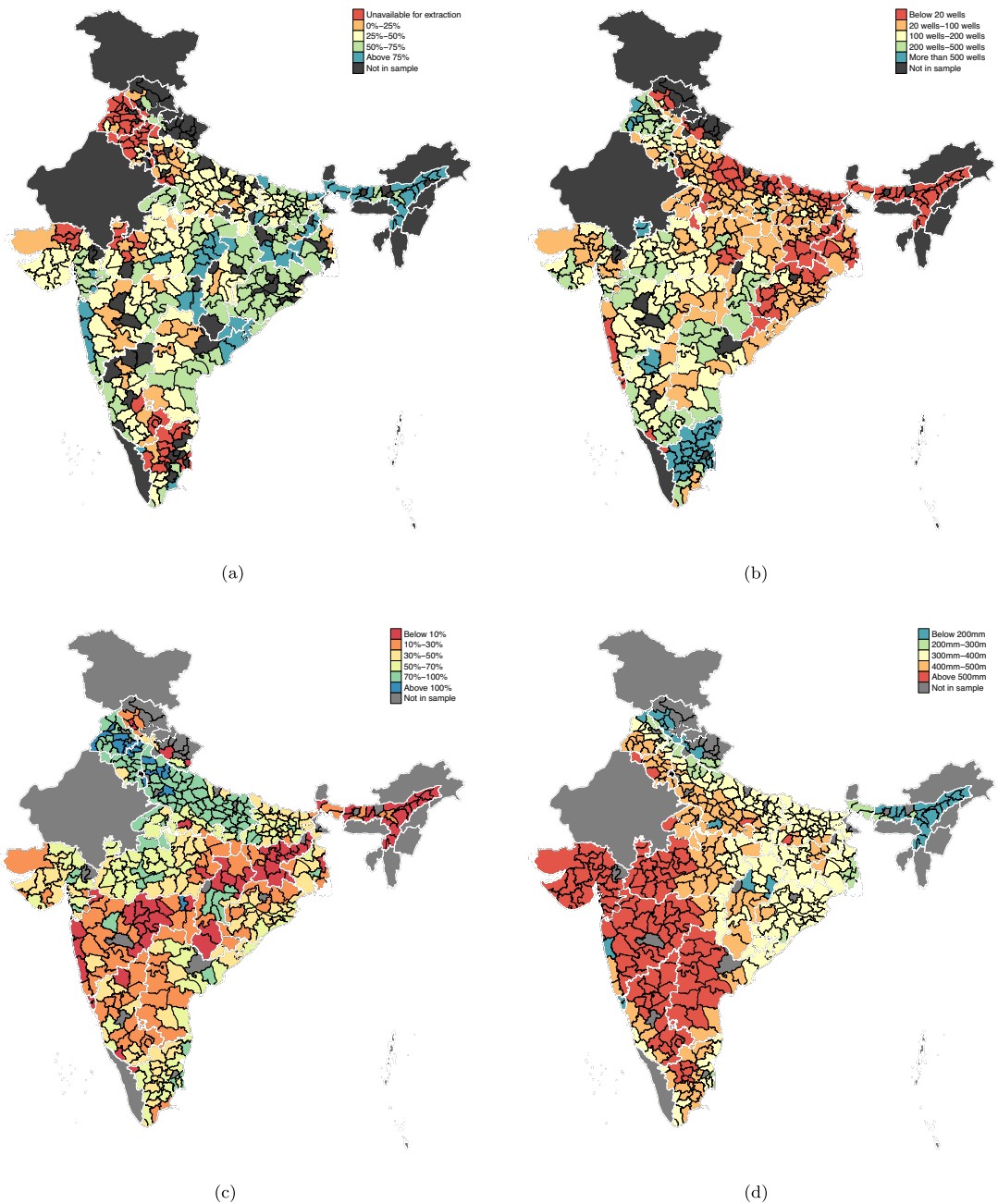

**Fig. 1 | District-level averages of groundwater and agricultures indicators.**
**a** Proportion of annually replenished groundwater available for extraction;
**b** number of electrified groundwater wells per 1000 hectares of cultivated area;
**c** proportion of cultivated area irrigated annually; **d** mean seasonal water deficit during winter cropping (November to March). Data on district-level mean monthly water deficit is averaged over 1958–2015, cultivated and irrigated area is averaged over 2005–2015[62]. Latest available data on district-level groundwater consumption is for 2017[66] and the number of electrified wells is for 2013–14[41].

The second wave of RE focused on domestic electrification and began a decade later in the early 2000s[13]. *Rajiv Gandhi Grameen Vidyutikaran Yojana* (RGGVY) launched in 2005, consolidated the various schemes on RE and provided 90% grant-based financing to states for universal household electrification. Along with universal electrification, RGGVY also provided households below the poverty line with free electricity connections. States were granted funding by the central government based on the number of rural households that needed electricity connections and prescribed electricity loads per household[38]. RGGVY has been credited with the rapid expansion of RE in India, with over 280M people connected to the electricity grid between 2000 and 2010[13]. Despite RGGVY's relative success in bridging the regional gap in household electrification (Fig. 3a, c), large

regional differences still remain in electrified pumping of groundwater (Fig. 3b, d).

## Results
### Household electrification had few spillovers to agricultural electricity after policy change
We created a novel dataset by matching multiple rounds of household census with groundwater well census to understand the impact of RE on the expansion of groundwater irrigation. Our final data consists of three main categories—groundwater well data, household demographics, and rainfall data at the district level spanning between 1986 to 2013. We quantify the divergence in the relationship between household electrification and use of electricity for agriculture arising

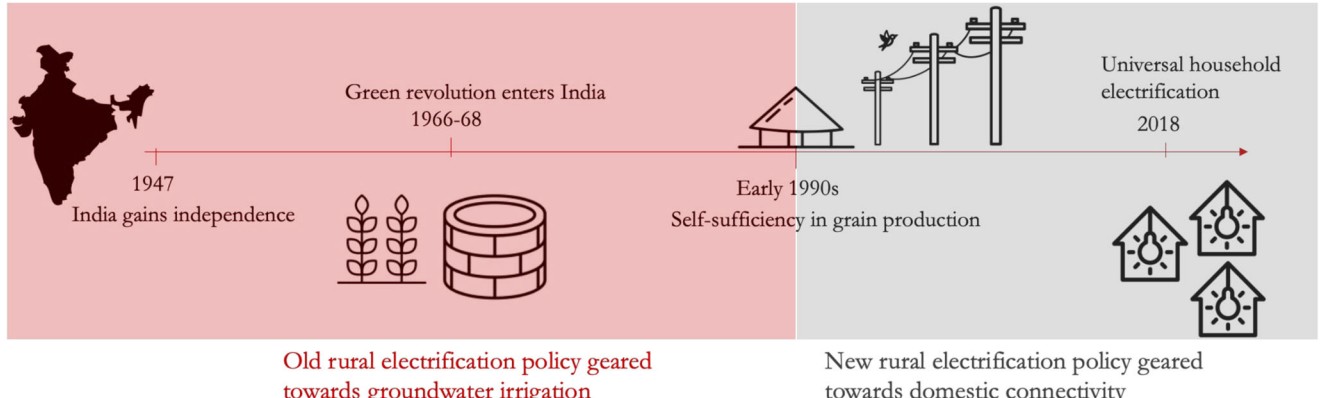

**Fig. 2 | Milestones in Rural electrification (RE) expansion in India during 1947–2018.** Data are based on India's five-year plans (1–7) chapters on irrigation and flood control.

from changes in RE policy focus. We do so by distinguishing between districts that were electrified during irrigation-focused RE policy from districts that were majorly electrified post a shift towards domestic consumption. We call the latter policy change districts denoted by *PC*. The median electrification rate across all districts was ~50% in 2001 and serves as the cutoff for classifying *PC*—i.e., districts that had less than 50% electrified households are coded 1 and those that had 50% or greater electrification rates in 2001 are coded 0. We also consider models where we include electrification quantiles instead of a single binary classifier and find similar results (see Supplementary Tables 2 and 3).

Before presenting our regression models, Fig. 4 presents cross-sectional, descriptive evidence to motivate our analysis. In 2013, our most recent round of data, we see a stark difference in the relationship between household electrification and electrified groundwater pumps in PC versus non-PC districts. PC districts had far fewer wells with electric pumps compared to non-PC districts with similar numbers of electrified households (Fig. 4). This divergent cross-sectional relationship between the two sets of districts could be driven by unobserved factors. For instance, groundwater endowments, agricultural suitability, and other similar factors could have affected both the timing of household electrification and the presence of electrified pumps within a district. To address such potential confounding variables, we use a two-way fixed effects regression strategy with a number of additional controls (see "Methods"). We estimate the impact of electrification on the number of groundwater wells with electric pumps and the area irrigated by groundwater for PC and non-PC districts.

Our fixed effects regression estimates measures the association between the number of electrified households with the number of groundwater wells using electric and diesel pumps across 1986 to 2013. The estimates show that, on average, the electrification of 100 additional households is associated with approximately 2 additional electrified wells among PC districts compared to 16 additional electrified wells among non-PC districts (Table 1). Results from the estimation of wells with diesel pumps add credibility to our findings. Diesel being far more expensive to operate (and limiting in terms of depths at which it can be used), is almost always a second choice to electricity-based pumping across India. Therefore household electrification is expected to be associated with a decrease in the number of wells with diesel pumps. We find this to be true for non-PC districts. However, we find no similar replacement occurring among PC districts. Our results imply that across PC districts, on average, household electrification is, in fact, associated with a small increase in the number of wells with diesel pumps. This increase could be due to a reallocation of household expenditure freed up from lighting and other needs met by domestic electrification.

We use districts selected in the first phase of electrification expansion carried out under RGGVY as an alternate way of identifying districts electrified after RE policy focus shifted from agriculture towards domestic electrification (Table 1 columns 5–7). We find qualitatively similar results with some loss in precision due to data limitations. To further test the robustness of our hypothesis, we also run a similar analysis for blocks (administrative unit between districts and villages) in Madhya Pradesh using the same definition of PC and non-PC districts to account for the large variations in agriculture across Indian states in terms of crops cultivated, agricultural returns and inputs used. Results from the block-level analysis are consistent with our main results, wherein electrification across blocks in PC districts are associated with a much smaller increase in electrified wells than that experienced by blocks among non-PC districts (results reported in Supplementary Figure 1 and Supplementary Table 1).

We find similar differences between PC and non-PC districts for areas irrigated by groundwater wells across cropping seasons (Table 2). In India, there are two main cropping seasons—*Kharif* coincides with the monsoons and *Rabi* with dry winter. Groundwater irrigation is particularly relevant during *Rabi*, but changing rainfall patterns have made groundwater irrigation important even during *Kharif* in recent years[39]. Our estimates indicate increases in irrigated area across all seasons among non-PC districts on average. However, we find a large penalty associated with electrification post policy change. Particularly for *Rabi*, the negative penalty of electrification post policy change is so large that we find no increases in irrigated area associated with household electrification among PC districts (column 5). Table 2 also reports results from specifications where we include the number of electrified wells (columns 2, 4, and 6). As expected, we find the number of electrified wells explains much of the variation in groundwater-irrigated areas across all districts and all seasons.

## Income does not explain irrigation differences between PC and non-PC districts

RGGVY removed upfront costs of domestic connections for households below the poverty line. Removal of the one-time connection costs could have led to a jump in the rates of domestic electrification with no similar increases in electricity connections for groundwater irrigation. Second, transformer sizing in RGGVY is based on aggregation of the number of households to be electrified and load capacity per household[40]. The highest prescribed load capacity of a single household at 500 W is incapable of running even a 1 HP electric pump (~745 W)[38]. For comparison, less than 3% of all wells excluding deep tube-wells were operated with pumps of capacities below 2 HP in 2013[41]. Deep tube-wells have depths greater than 70 m and require pumps with capacities greater than 10 HP (see Supplementary Note 1).

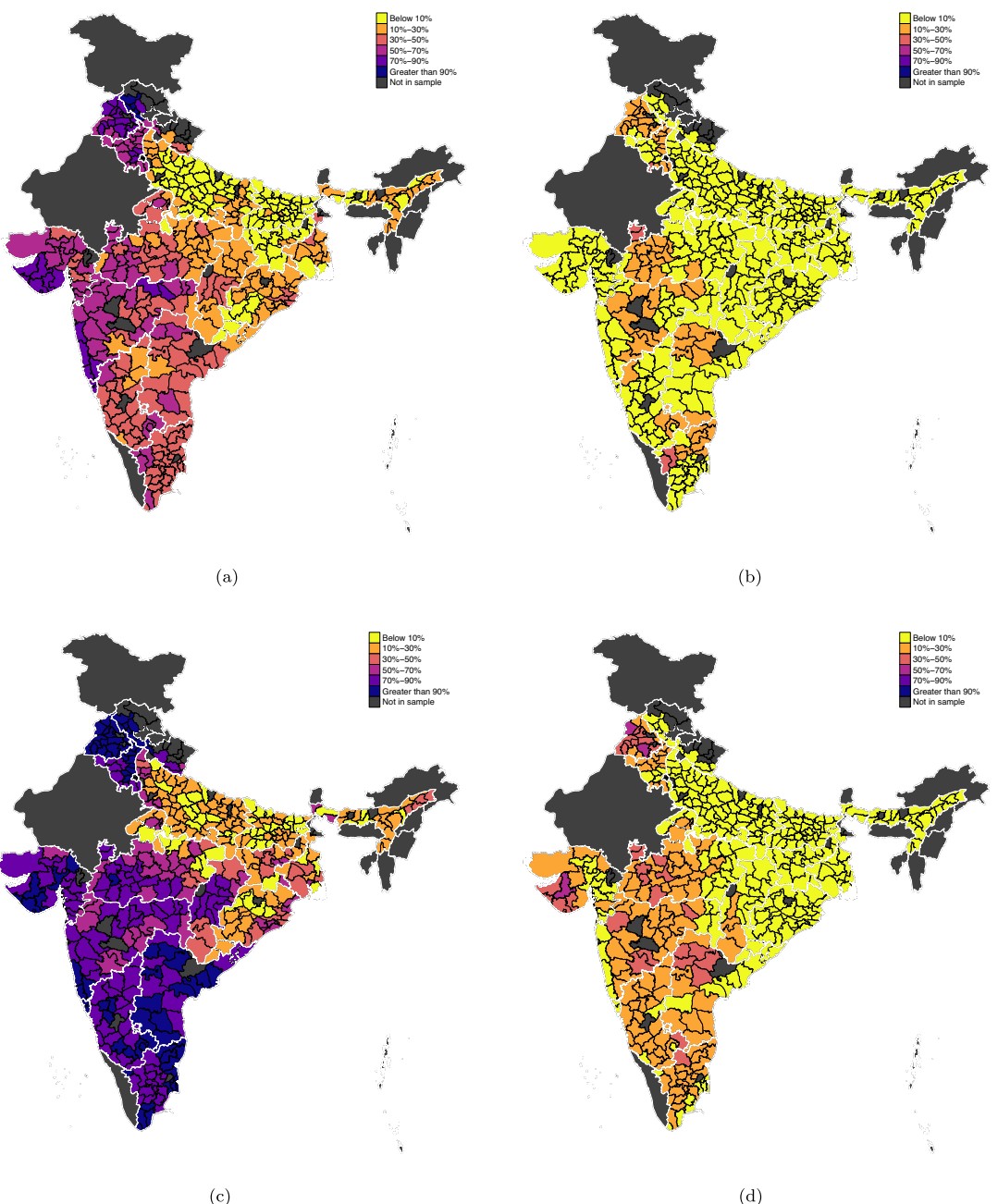

**Fig. 3 | District-level averages of electrified households and electrified irrigation wells.** District-level shares of: **a** electrified households in 1986; **b** households with electrified wells in 1986; **c** electrified households in 2013; **d** households with electrified wells in 2013. Data are based on district-level panel dataset ($n = 323$). Gray lines represent district boundaries, and white lines represent state boundaries.

Therefore, in PC districts where electrification majorly occurred under the RGGVY program, transformer capacities could be constraining even households with the financial means to access electricity for groundwater irrigation[40,42]. A second set of analyses using household-level data allows us to test whether economic constraints (household wealth) contributed to the observed reduction in groundwater irrigation among PC districts.

To test the relationship between household wealth and groundwater irrigation, we compare area irrigated during 2013 by farming households of similar wealth across PC and non-PC districts. We use the latest government survey of farming households which is representative at the district-level and covers all states. Household wealth is measured by consumption percentiles which we construct based on all surveyed farming households (see Supplementary Table 5). Overall we find a stronger wealth gradient in *Rabi* irrigated area compared to

*Kharif* irrigated area among non-PC districts (Fig. 5, Supplementary Table 6).

Our regression model omits the lowest consumption percentile (15th percentile) from non-PC districts to avoid multi-collinearity. Figure 5 presents the difference in irrigated area for the remaining consumption percentiles relative to this reference category. Compared to the poorest 15th percentile of households, all other farming households in non-PC districts irrigated larger amounts of cultivated land during *Rabi*. However, no similar difference in irrigated area was found between the poorest 15th percentile households in non-PC districts and households among PC districts that were even in the top 85th consumption percentile.

No wealth gradient was detected for *Kharif* irrigated area and is perhaps reflective of the non-essentiality of irrigation during *Kharif* due to the monsoons. A point to note here—data limitations prevent us

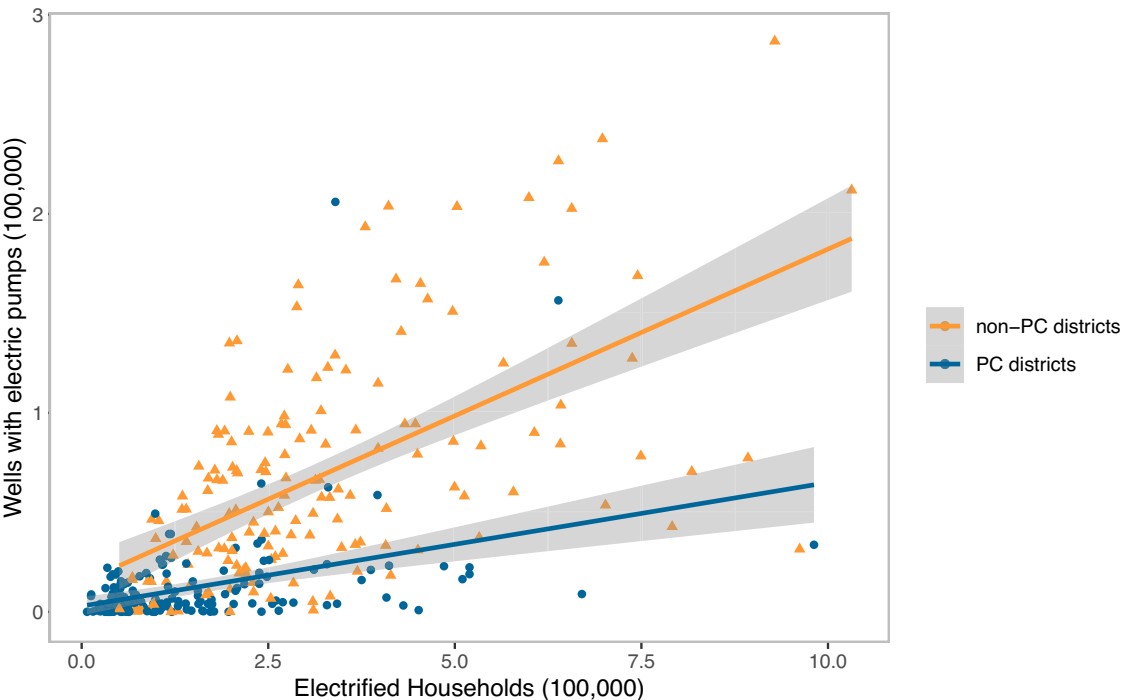

**Fig. 4 | Relationship between electrified households and electrified wells in 2013 for non-Policy Change (non-PC) and Policy Change (PC) districts.** PC is a binary indicator for districts that were majorly electrified after policy change ("Methods"). The point estimates represent district-level data on electrified houses and wells with electrified pumps. The lines represent linear best-fit models with shaded 95% confidence bands. Data are based on district-level population and minor irrigation census data (*n* = 323).

from making any distinction between the different irrigation sources in this estimation, but it is safe to assume that irrigated area majorly implies groundwater irrigation at least during *Rabi*. Surface water irrigation is useful only during the monsoons and could be one of the reasons that explains the missing gradient during Kharif—gravity flow-based irrigation from canals and other surface storage offer little control to farmers and are dependent on the amount of rainfall received in that year[43]. Therefore individual household wealth likely has a small role to play in irrigated area during Kharif.

Our regression model included an extensive set of control variables that could impact the area irrigated by a household (see "Methods"). Results in Fig. 5 use the broadest definition of consumption. To check for robustness of our results, we ran the model with alternative definitions of consumption and found similar results for even the most parsimonious definition of consumption (Supplementary Table 7).

Absence of any wealth gradient in *Rabi* irrigated area negates possible concentration of poverty as a reason for the loss in electrified groundwater wells and irrigated area among PC districts. Instead, our results imply that farmers in PC districts are unable to physically access electricity for groundwater irrigation. Physical access to electricity could mean poor electricity supply related to duration of supply, voltage fluctuations, brownouts, and blackouts. It could also mean that farmers don't have an electric connection to allow the flow of electrons to power their pump-sets in the first place. Unfortunately, data on transformer sizing and electricity supply do not exist for agriculture at the district level in India to identify which type of physical access to electricity constraints groundwater irrigation. However, two independent evaluations of RGGVY found both transformer failure and limited transformer capacities to be major drivers of poor quality of electricity experienced by RGGVY electrified regions[40,42].

## Discussion
We find that despite the ubiquitous presence of electricity infrastructure across rural India, the use of electricity for groundwater irrigation remains regionally concentrated. Regions with electric pumping of groundwater are those that were electrified before RE policy was reconfigured towards domestic electrification. Our results imply large absences in the use of electricity in groundwater irrigation among PC districts. The effect size of −0.1362 translates to an average loss of over 20,000 electrified groundwater wells across PC districts. In 2013, PC districts on average had little over 12,000 electrified wells. The average loss in irrigated area amounts to nearly 16,000 hectares during *Rabi*, implying the loss of over one-fifth of *Rabi* irrigated area in 2013 among PC districts attributable to the policy change. Our analysis does not exhaustively capture all possible disincentives to electric pump investments—poor supply of electricity, small or scattered parcels of landholdings and inadequate returns to irrigation are some potential biases to our results. For instance, households on average owned smaller parcels of land among PC districts compared to non-PC districts. While we control for land sizes in our analysis, we are unable to consider poor quality of supply, scattered parcels or returns to irrigation due to data limitations. We are also unable to account for water markets that are reported to exist in some eastern states in the country[44]. Nevertheless, these factors are important to account for while designing policies that target irrigation expansion across PC districts.

Across the developing world, the impact of irrigation on poverty is found to be positive on average, although empirical evidence suggests wide ranges in the sizes and sectors of the positive impacts. The latter include farm and non-farm employment, wages, food prices, production volumes, and nutritional outcomes[45–47]. In India, loss in groundwater access has led to dramatic shrinkage in agricultural incomes[48]. Groundwater depletion also threatens the extent to which farmers can use irrigation to offset production losses from rainfall variability[49]. Expanding electrified groundwater irrigation is important not only to protect current food production but to also shift towards a more sustainable paradigm of groundwater consumption. Large parts of regions electrified post policy change are located in eastern India,

**Table 1 | Association between late-electrification and energy source for groundwater pumps**

| | Groundwater wells | | | | | |
| --- | --- | --- | --- | --- | --- | --- |
| | Using *PC* | | | Using *RGGVY* | | |
| | 1 | 2 | 3 | 4 | 5 | 6 |
| | Total | Electric pumps | Diesel pumps | Total | Electric pumps | Diesel pumps |
| Electrified households | 0.092*** | 0.158*** | −0.039*** | 0.067*** | 0.149*** | −0.042*** |
| | (0.021) | (0.020) | (0.008) | (0.025) | (0.023) | (0.010) |
| Electrified households × *PC* | −0.086*** | −0.136*** | 0.056*** | | | |
| | (0.021) | (0.022) | (0.014) | | | |
| Electrified households × *RGGVY* | | | | 0.005 | −0.054** | 0.034*** |
| | | | | (0.025) | (0.025) | (0.011) |
| Total households | 0.029** | −0.013* | 0.002 | 0.0152 | −0.032*** | 0.010 |
| | (0.014) | (0.007) | (0.018) | (0.013) | (0.010) | (0.015) |
| **Fixed-effects** | | | | | | |
| District | Yes | Yes | Yes | Yes | Yes | Yes |
| Year | Yes | Yes | Yes | Yes | Yes | Yes |
| **Fit statistics** | | | | | | |
| Observations | 969 | 969 | 969 | 945 | 945 | 945 |
| R² | 0.72 | 0.90 | 0.80 | 0.71 | 0.89 | 0.79 |
| Within R² | 0.07 | 0.53 | 0.11 | 0.02 | 0.21 | 0.03 |

Each column presents results from a linear regression model of the number of groundwater wells with pumps on the number of electrified households estimated using equation (1). Data consists of a district-level panel dataset constructed from district-level population and minor irrigation census data. In columns (1)–(3), late-electrification is measured using *PC*—a binary indicator for districts that were majorly electrified after policy change. In columns (4)–(6), use an alternate measure of late-electrification using *RGGVY*—a binary indicator for districts selected during phase I of RGGVY (see "Methods"). Standard errors are shown in parentheses and are clustered at the district-level. Stars denote statistical significance of the coefficients: *$p < 0.1$, **$p < 0.05$, and ***$p < 0.01$, respectively.

**Table 2 | Association between late-electrification and expansion in annual and season-wise area irrigated**

| | Irrigated area (ha) | | | | | |
| --- | --- | --- | --- | --- | --- | --- |
| | Annual (ha) | | *Kharif* (ha) | | *Rabi* (ha) | |
| | (1) | (2) | (3) | (4) | (5) | (6) |
| Electrified households | 0.140*** | −0.043 | 0.105*** | −0.006 | 0.086*** | 0.010 |
| | (0.050) | (0.043) | (0.025) | (0.022) | (0.020) | (0.020) |
| Electrified households × *PC* | −0.207*** | −0.051 | -0.100*** | −0.005 | −0.108*** | −0.041 |
| | (0.0608) | (0.053) | (0.033) | (0.027) | (0.029) | (0.026) |
| Wells with electric pumps | | 1.158*** | | 0.710*** | | 0.483*** |
| | | (0.159) | | (0.110) | | (0.073) |
| Total households | 0.063 | 0.076* | 0.015 | 0.023 | −0.020 | −0.014 |
| | (0.044) | (0.043) | (0.018) | (0.018) | (0.020) | (0.020) |
| Cumulative average monthly rain | −1.301 | −5.225 | −7.970** | −10.99*** | 43.57 | 47.10* |
| | (6.134) | (5.562) | (3.126) | (2.924) | (27.48) | (25.63) |
| **Fixed-effects** | | | | | | |
| District | Yes | Yes | Yes | Yes | Yes | Yes |
| Year | Yes | Yes | Yes | Yes | Yes | Yes |
| **Fit statistics** | | | | | | |
| Observations | 969 | 969 | 969 | 969 | 969 | 969 |
| R² | 0.86 | 0.87 | 0.85 | 0.87 | 0.85 | 0.86 |
| Within R² | 0.34 | 0.39 | 0.32 | 0.41 | 0.24 | 0.28 |

Each column presents the results of a linear regression of the area irrigated by groundwater wells on the number of electrified households estimated using equation (1). Data consists of a district-level panel dataset constructed from district-level population and minor irrigation census data. Late-electrification is measured using *PC*—a binary indicator for districts that were majorly electrified after policy change (see "Methods"). The dependent variable in columns (1)–(2) is the annual irrigated area. Columns (3)–(4) show the results for the *kharif* (monsoon) cropping season, while columns (5)–(6) show the results for *rabi* (winter) cropping season. Cumulative average monthly rainfall is calculated based on annual rainfall for columns (1)–(2), rainfall from November to March for columns (3)–(4), and rainfall from June to October for columns (5)–(6). Standard errors are shown in parentheses and are clustered at the district-level. Stars denote statistical significance of the coefficients: *$p < 0.1$, **$p < 0.05$, and ***$p < 0.01$, respectively.

where relatively healthy groundwater levels, low cropping intensities, and lower irrigation requirements make the region a prime candidate for the future bread basket of the country[28,50,51]. However, cultivation expansion in the east is unlikely to occur with current levels of irrigation[39]. Therefore our results offer an avenue to increase *Rabi* cultivation in the east by fixing the electricity infrastructure to serve groundwater irrigation and safeguard India's future food security.

Providing irrigation access alone will not spur *Rabi* cultivation among PC districts. Ensuring well-functioning agricultural produce markets will be key and also requires policy interventions as government procurement dominates the Indian agricultural markets[50]. Agricultural policy is determined by individual state governments which varies government procurement of agricultural produce across state boundaries[52,53]. Madhya Pradesh is among the largest procurers of wheat, majorly grown during *Rabi*[54]. Yet, we find that nearly a third of its districts electrified post policy change and have far fewer electrified groundwater wells, *Rabi* irrigated area, and wheat production (Supplementary Table 8). Therefore, providing electricity access for groundwater irrigation, while not sufficient, is certainly necessary to spur *Rabi* cultivation.

Our results also highlight the role of physical constraints in electricity in limiting groundwater irrigation among districts that majorly electrified after policy change. The absence of any wealth gradient in *Rabi* irrigated area among PC districts is in stark contrast to the positive wealth gradient we find in non-PC districts. Therefore physical access rather than economic access to electricity is instrumental in limiting groundwater irrigation in regions electrified post policy

change. In line with our results, no meaningful increases in the number of irrigation wells or proportion of area cultivated or irrigated in RGGVY electrified villages was found in earlier studies[1]. Our results therefore spell the need to revisit the current definition of electricity access measured solely by household electrification. Just as the Indian government's focus on household electrification has curtailed the use of electricity in groundwater irrigation, it may also be constraining other parts of the rural economy. The target of electricity access programs is household electrification in most developing countries. Perhaps it is time for more studies to isolate the different avenues whereby electricity impacts rural economies across the developing world and maximize electrification's role in poverty alleviation.

## Methods
### Data source
We create a novel district-level panel dataset spanning the period from the mid-1980s to the 2010s by matching four sources of data published by the Government of India. Our central analysis relies on multiple rounds of the (i) Population and household census of India and (ii) Minor Irrigation Census. We supplement these district-level data with household information from the Situation Assessment Survey of Agricultural Households collected by the National Sample Survey Organization (NSSO) in 2013. Finally, our data also includes district-

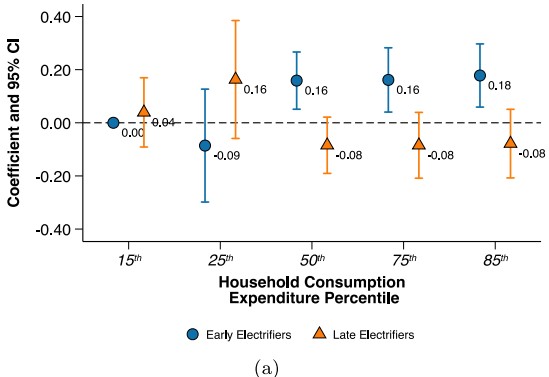
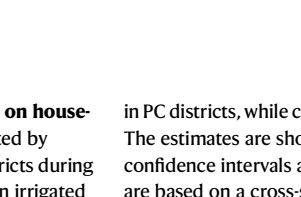
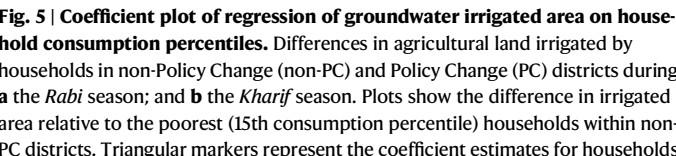

**Fig. 5 | Coefficient plot of regression of groundwater irrigated area on household consumption percentiles.** Differences in agricultural land irrigated by households in non-Policy Change (non-PC) and Policy Change (PC) districts during **a** the *Rabi* season; and **b** the *Kharif* season. Plots show the difference in irrigated area relative to the poorest (15th consumption percentile) households within non-PC districts. Triangular markers represent the coefficient estimates for households in PC districts, while circular markers correspond to households in non-PC districts. The estimates are shown by the numbers beside the markers. Whiskers show 95% confidence intervals accounting for district-level clustering. Coefficient estimates are based on a cross-sectional regression using nationally representative survey data of agricultural households ($N = 11,182$) in 2012–13 ("Methods"). PC is a binary indicator for districts that were majorly electrified after policy change ("Methods").

level measures of rainfall from the Indian Meteorological Department, groundwater consumption data from the Central Groundwater Board, and water-deficit, cultivated, and irrigated areas from the district-level database compiled by ICRISAT-TCI[55–62].

**Population census of India.** India conducts decennial household censuses. We use digitized data from the last three waves conducted in 1991, 2001, and 2011. In each census wave, the key variables include the absolute number of electrified rural households and total rural households. We define RE as the number of electrified rural households, similar to the current definition of RE by the Government of India. We match districts across the three waves of the household census, in addition to matching districts in each wave of the household census to each wave in the minor irrigation census.

**Minor Irrigation census.** The Ministry of Water Resources under the Government of India collects Minor Irrigation (MI) census data every 5 years since 1986. We use the first, third, and fifth waves of the census carried out during 1986–87, 2000–01, and 2013–14, respectively. Gujarat and Maharashtra, two states with large numbers of groundwater wells are missing in the second wave carried out during 1993–94. The fourth wave carried out during 2006–07 falls in between the two household census waves of 2001 and 2011. For these reasons, we exclude both the second and fourth waves of the minor irrigation census from our analyses. The MI census provides details of all irrigation structures that irrigate less than 2000 hectares[20]. These structures are categorized into three types of groundwater wells and two types of surface water irrigation infrastructures. We use data on the three types of groundwater wells, which include dug wells, shallow tube-wells, and deep tube-wells. Dug wells are defined as wells that are constructed without the use of a boring (drilling) machine and are relatively shallow, ranging from 8–15 m. Shallow and deep tube-wells are both constructed using drilling machines and differ from one another in their depths—shallow wells extend to 70 m, and deep wells are classified as wells with depths greater than 70 m. The 5th MI census wave further categorizes shallow tube-wells into shallow (less than 35 m deep) and medium tube-wells (35–70 m deep). We collapse both categories into shallow tube-wells for consistency across years. Surface water irrigation accounts for a little over 10% of the total area irrigated by MI structures, and is not included in our analysis.

**Matching districts across population and MI census.** New districts are formed in India for many reasons, including the creation of new states and population increase (although there exists no uniform

benchmark for population or population density for the formation of new districts). Our analysis uses the 1986 MI census to map districts consistently across all MI and population census rounds. Where possible, we merge newly formed districts into their original 1986 boundaries. We drop districts whose boundaries could not be consistently matched—for instance, in cases where a district was formed by partially combining areas from more than one previously existing district. We also drop districts for which the MI census data were missing for any of the waves. In all, our data consists of 323 districts across 18 states in 1986.

The years in which the MI census was conducted do not precisely match the household census's years. However, our assumptions lead to a conservative estimate of the difference in the impact of electrification on groundwater irrigation pre and post-2001. It is likely that fewer houses were actually electrified in 1986 in the non-PC districts than the reported number in the 1991 household census, leading to a downward bias in the impact on groundwater irrigation among non-PC districts. Following a similar logic, it is likely that more households were actually electrified in 2013 in the PC districts (compared to the number reported in the 2011 population census), with a possible upward bias in the impact on groundwater irrigation among PC districts. Therefore the estimated overall difference in the impact of electrification on groundwater irrigation is expected to be downward biased and represents a conservative estimate of the true impact.

**Situation Assessment Survey of Agricultural Households.** The National Sample Survey Office (NSSO) operates under the Ministry of Statistics and Program Implementation, Government of India. In its 70th round of survey in 2013, the NSSO collected information from agricultural households on consumption, farming practices, resource availability and awareness about various government-supported agricultural schemes. The NSSO defined an agricultural household as one producing a cumulative annual value of INR 3000 or more from agricultural activities and with at least one household member self-employed in agriculture (either as a primary or secondary income-generating activity). The survey was carried out during two visits—the first visit overlapped with the *Kharif* season, and data was collected for July to December 2012 and the second visit roughly overlapped *Rabi*, and data was collected for January to June 2013. The main variables of interest from this dataset are per capita consumption in the last 30 days, area irrigated and cultivated during the two cropping seasons and the total land operated in a year (both owned and leased) by a household. We construct consumption percentiles across agricultural households to tease out the role of wealth in accessing irrigation. We

use a subset of the total data and only include households that identify cultivation as their primary source of income to reduce any bias from the impact of income from other sectors on the ability to irrigate. Once again, we match the districts from MI and household censuses to identify PC districts, which allows us to compare irrigated area among households of similar consumption percentiles in PC and non-PC districts.

**Rainfall data.** There are ~3500 meteorological stations spread across 36 meteorological subdivisions in India. The Indian Meteorological Department (IMD) publishes monthly data at the district and sub-division levels. We use district-level data for the same 3 years as the Minor Irrigation Census—1986, 2000, and 2013. We impute missing data by substituting them with sub-divisional data for the few years and districts that were missing.

**Rajeev Gandhi Grameen Vidyutikaran Yojana (RGGVY) phase I district selection.** RGGVY was carried out under India's *X*th (2002–2007), *XI*th (2007–2012) and *XII*th (2012–2017) 5-year plans. We use districts selected during phase I (2002–2007) as an alternate method to identify districts which were majorly electrified after the rural electrification policy shifted towards domestic electrification[1].

**Night time luminosity.** The Defense Meteorological Satellite Program (DMSP) Operation Linescan System (OLS) collects global images twice per day. We use archival annual measures of night time luminosity which has been matched to Indian villages from 1994 to 2018[63]. We use night time luminosity as an additional source as the household census does not publish electrification data at the block level. We use total light luminosity values which range from 0 to 63 which has been calibrated for consistent measure across the range of years 1994–2013[64]. The total luminosity captures the brightness of all pixels located within each block.

The first minor irrigation census is not available at the village or block levels, and the earliest year of observations for the DMSP-OLS is 1994. The second minor irrigation census was carried out during 1993–94. We matched the second, third, and fifth rounds Minor Irrigation Census with annual night time luminosity measures for 1994, 2000, and 2013 respectively for each block in Madhya Pradesh. We were able to match 235 of the 377 blocks reported in the Minor Irrigation Census.

## Estimation strategy

We use a two-way fixed effects estimation strategy to study the impact of RE on groundwater irrigation across PC and non-PC districts. Tables 1 and 2 report results from model (1).

$$Y_{it} = \alpha \, \text{elechh}_{it} + \beta \, \text{elechh}_{it} \times PC_i + \delta \mathbf{X}_{it} + \mu_i + \gamma_t + \epsilon_{it} \qquad (1)$$

The outcome variable $Y_{it}$ is the number of groundwater wells (in Table 1 and the area irrigated by groundwater wells (in Table 2) for district $i$, at time $t$. These outcomes are measured using data from the Minor Irrigation censuses.

The continuous variable $\text{elechh}_{it}$ is the number of electrified rural households in district $i$ and time $t$ from the Population Census data. $PC_i$ is a binary treatment variable that takes the value 1 for districts that had less than 50% of rural households electrified by 2001.

The coefficient on the interaction of $\text{elechh}_{it}$ with $PC_i$, $\beta$, is of main interest. It captures the difference in the effect of household electrification on pump electrification between PC and non-PC districts. In other words, $\alpha$ captures the relationship between rural electrification and groundwater irrigation among non-PC districts, while $\alpha + \beta$ captures the estimate for PC districts. We also use an alternate measure of PC districts as those selected under phase I of rural electrification expansion under RGGVY.

The control variables, $X_{it}$, include the total number of rural households for when we measure the impact on the number of groundwater wells. $X_{it}$ includes both the total number of rural households and the cumulative monthly rain when we measure the impact on groundwater-irrigated area. The cumulative monthly rain includes only the specific months of cultivation. We do not include cumulative monthly rain while estimating the impact on groundwater wells, as we assume the decision to construct wells and buy pump-sets to be long-term investments not impacted by yearly variations in rainfall. $\mu_i$ denotes the full set of district-level fixed effects to control for time-invariant differences between districts. $\gamma_t$ denotes year fixed effects to control for aggregate time shocks common across all districts. Standard errors are clustered at the district level. For an unbiased estimate, there should be no time-varying district-level factors that are correlated with both extent of household electrification and groundwater irrigation. While this assumption cannot be tested empirically, our set of time-varying controls account for the main sources of such bias. As a robustness test, the estimates on groundwater irrigation remain qualitatively similar when we exclude rainfall controls, suggesting that the results are unlikely to be driven by district-level time-varying factors (see Supplementary Table 4).

Our main results use less than 50% of rural households electrified in 2001 as the threshold to define $PC_i$. Our results remain similar when we define multiple stages of electrification instead of a binary categorization. In Supplementary Tables 2 and 3, we define rural electrification coverage dummies based on 15th, 25th, 50th, 75th, and 85th percentiles of the electrification rate. We see that PC districts (i.e., districts that were in lower percentiles of electrification coverage distribution in 2001) see a smaller number of electric pumps for each additional household electrified.

We use the same definition of PC and non-PC districts in the block-level analysis for Madhya Pradesh. However, we measure electrification using night time luminosity as household electrification data are unavailable at the block-level. Supplementary Table 1 report results from model (2).

$$Y_{bit} = \alpha \, \log(\text{nl}_{bit}) + \beta \, \log(\text{nl}_{bit}) \times PC_i + \delta \mathbf{X}_{it} + \mu_i + \gamma_t + \epsilon_{it} \qquad (2)$$

Here the outcome variables $Y_{it}$ are the number of groundwater wells powered by different pumping energy sources for block $b$ in district $i$, at time $t$. These outcomes are measured using data from the Minor Irrigation censuses. The continuous variable $\log(\text{nl}_{bit})$ is the log of the total night time luminosity in block $b$ in district $i$ and time $t$ from DMSP-OLS. $PC_i$ is the same binary treatment variable as in model (1) and takes the value 1 for districts that had less than 50% of rural households electrified by 2001. The coefficient on the interaction of $\log(\text{nl}_{bit})$ with $PC_i$, $\beta$, is of main interest. It captures the difference in the effect of electrification measured by night time luminosity on pump electrification between blocks in PC and non-PC districts. The control variable, $X_{it}$, includes the total number of villages. $\gamma_t$ denotes year fixed effects to control for aggregate time shocks common across all districts. Standard errors are clustered at the district level.

In the second set of analyses using household-level data, we compare seasonal irrigated areas in 2013 between households in pre and post policy change which are in the same wealth percentiles. Figure 5 reports results from the regression:

$$Y_{hi} = \alpha \, \text{ConsPerc}_{hi} + \beta \, \text{ConsPerc}_{hi} \times PC_i + \delta \mathbf{X}_{hi} + \gamma \text{Rain}_i + \text{State}_i + \epsilon_{hi}$$
$$(3)$$

where, $Y_{hi}$ is irrigated area for household $h$ in district $i$. $\text{ConsPerc}_{hi}$ is the wealth indicator measured by the per capita consumption percentile of household $h$ in district $i$, and $PC_i$ is 1 for districts electrifying post policy change and 0 for pre-policy change districts, the same binary variable used in model (1). $X_{hi}$ is a vector of control variables at the household

level and includes the total land cultivated by the household and other household and head of household controls such as household size, expenditure on non-farm activities and education and agricultural training of the household head. $Rain_i$ is the cumulative average monthly rainfall in district $i$ from January to June 2013, when the survey was conducted for *Rabi* and July to December 2012 for Kharif. Since agricultural policy is a state subject in India, we include $State_i$ to control for state-level differences in agricultural policy, which could impact irrigated areas. By including state fixed effects, our estimates effectively compare households across PC and non-PC districts within the same state. We cluster standard errors at the district level.

### Reporting summary

Further information on research design is available in the Nature Portfolio Reporting Summary linked to this article.

### Data availability

All data required for replication are publicly available[65] on Zenodo at https://doi.org/10.5281/zenodo.8430025.

### Code availability

All code required for replication are publicly available[65] on Zenodo at https://doi.org/10.5281/zenodo.8430025.

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

## Acknowledgements

S.R. was funded by Kimmelman Family Fellowship in the Emmett Interdisciplinary Program in Environment and Resources, and McGee Levorsen grant by the School of Earth, Energy and Environmental Sciences at Stanford University. The authors would like to thank Dr. Roz Naylor, Dr. Samantha Sekar, and Dr. Nina Brooks for their feedback and valuable suggestions.

## Author contributions

S.R. contributed the primary development of the research question and data collection. S.R. and H.P. jointly developed the analysis, conclusions and manuscript.

## Competing interests

The authors declare no competing interests.
