## [Peer Review File · Nature Communications]

Reviewers' Comments:

Reviewer #1:

Remarks to the Author:

The central premise of this project is that rural electrification impacts household incomes and a key pathway through which it does so is via enabling households engaged in agriculture to pump groundwater for irrigation. The hypothesis which they proceed to test is that a 2000s turn in policy focus towards household electrification rather than electrification for agriculture, which therefore undermined the growth of electric pumps in agriculture. This "crowding out" of electric agricultural pumps by domestic electrification in turn affected income growth. They proceed to demonstrate this through a series of steps – first, by demonstrating that the relationship of electricity based pumping and domestic access to electricity is on average lower among "late electrifier" districts and the "early electrifier" districts and by showing that rural electrification was not income enhancing for the "late electrifiers" as it was for the "early electrifiers".

This paper is on a very interesting topic and the authors make very original and creative use of the data. It is motivated very well and the authors produce very useful insights. However, I was left underwhelmed by their bewildering empirical approach and the confusing use of data.

The backbone of this paper is the empirical specification in Lines 319-322 that regresses electric pumps (and then acreage irrigated by them) on households with electricity and an interaction term that combines electrified households with an indicator variable denoting if this is greater than 50% of all households. The authors claim

The coefficient on the interaction of $\text{elec} \times \text{LateElec}$, β , is of main interest. It captures the difference θ in the effect of household electrification on pump electrification between early and late electrifying districts. In other words, α captures the relationship between rural electrification and groundwater irrigation among early electrifying districts, while $\alpha + \beta$ captures the estimate for late electrifying districts.

I beg to differ. In essence, this is like a classic structural break model but defined in terms of a 50% threshold based on electrified households. Note that this is exactly the specification one would use to detect threshold effects that beyond a threshold of electricity diffusion in the village, the relationship between domestic and agricultural use of electricity changes. One could tell several alternate stories with this model: that there is a preference ordering of investment among rural households so that in the early stages of diffusion of electricity people invest in electricity pumps; or a story of village level development – in the early stages of electrification, one invests in agriculture (electric pumps) and eventually as surplus is generated in agriculture, one invests in off farm activities. I simply don't see the empirical specification delivering the story that the authors are trying to get at. Indeed, one could argue just based on the model, that it has little to do with any policy or crowding out, but a story of the evolution or dynamics of electricity use.

The authors robustness checks "We also consider models where we include electrification quantiles instead of a single binary classifier and find similar results (see appendix tables A1 and A2), in fact illustrate the diminishing importance of agricultural pump electrification as domestic electrification expands. In short, the authors have an exciting story here (they should write that one), but it is not the one they are trying to tell!

The fundamental problem is "construct" validity. Their bewildering approach to identifying those districts "exposed" to the policy and irrigation oriented versus a domestic oriented electrification policy, named "early" and "late" electrifiers derives not from the time when electrification was introduced but to the extent of electrification (in 2001?) i.e., as "majorly electrified" (lines 116). I struggled to see for myself nor access evidence to see why this is a good construct of early versus late

and not more versus less!

A better approach would be a structural break in terms of time (treat 1986 as a base period and use an indicator variable for post 2000s) to proxy the policy shift. The authors could add a piece-wise analysis based on what proportion of households are electrified.

I would always want state-year interaction fixed effects to control for time-varying policy changes (including electricity pricing policies, irrigation technology subsidies, other infrastructure, etc.)

I would like to see a greater discussion of diesel pumps (lines 66-68) used not as a substitute for electricity but for poor quality of electricity and hours of power supply. There are several discussions of these in the literature on subsidies in India (notably by Ashok Gulati and co-authors).

Line 23 "Perhaps it is time to reevaluate uniform prescriptive electricity loads and uses in formulating RE policies" This sentence can be clarified. It becomes clear later on but on line 23, it is too early for readers unfamiliar with the policy and the sector.

The authors claim that they use "newly constructed panel-dataset spanning three decades" (in the abstract). Yet the figure 2 notes that the "Relationship between electrified households and electrified wells in 2013 was stronger among early electrifiers compared to late electrifying districts. The point estimates represent district-level data on electrified houses and wells with electrified pumps." This suggests misleadingly that the regression was run for just 1 year, i.e., 2013.

The undertone at least in the initial parts of the paper is a celebration of electricity for pumping groundwater, barring some reflections in line 68. Perhaps you could start the paper early on by suggesting that despite the problem of overextraction etc. irrigation is known to have large benefits, and the culprit for that is more in the pricing than the extraction.

Reviewer #2:

Remarks to the Author:

The authors tackle a very important question with both energy and human development implications.

This is a good dataset and strong analytics – but the interpretations and framings need to be questioned. The study very nicely shows there is a disconnect for irrigation coverage vs. household electrification by time (early vs late), but it's not clear causality can be linked to policies favoring household electrification.

Utilities likely did slow down connections to farmers, but that was independent of any focus on household electrification. Farmers underpay, as well known, and this non-remunerative supply is a strain on utilities. Note, free power to farmers only started in 1977 in Andhra Pradesh, and wasn't widespread for a few more years.

While electricity access policy DOES favor households, it's not clear why that would be at the expense of agriculture. BOTH segments enjoy heavily cross-subsidized and subsidized electricity (cross-subsides for homes being for lower slabs or tiers of consumption).

One has to start with the reality that agricultural supply is very expensive for the utility – being both subsidized and cross-subsidized. Are there enough takers for a pumpset? Let's also start with SECC data – 55% of farmers are landless (laborers) so increasing pumpsets helps a subset. Here's a counterfactual Q: Now that households are ~100% electrified, does this now "free up" policy for pumpsets? The pumpsets policy has been the way it is (limited by design, with largesse and political

connections important) for a long while in the face of heterogeneity in connectivity.

There is no limited incorporation of the physics, design, and practicality of rural connections. Recent policies (last ~10 years) have been towards feeder segregation, but even before this, most states had "rosters" or schedules where agricultural supply was limited to, say, 6 or 8 hours per day. This was controlled through phase-separation, with single phase supply (for homes) meant to be as much as possible, ideally 24 hours in theory (but never in practice).

The second reality that questions the model and framing is how we have "incremental electrification". Most challenges for household electrification have been with the "last mile connection". The earlier (and insufficient) definitions of electrification focused on a single lightbulb meant the village was electrified, updated to then be 10% of homes. This was progressively upgraded, which is a good thing, but there is no evidence of a *policy reason* this was at the expense of pumpsets. Your analysis does bring out the point that household supply doesn't increase wealth much – so it is a separate question of how much one can/should increase pumpsets. If they were charged "full cost" there would be no problem but we know they are not. Money for loss-making utilities is scarce, and so there is a call to be made how much pumpset deployment is appropriate.

Aside: There was also a period (early 2000s) where power supply quality impacted affordability of pumpsets. Even with "cheap power", frequent burnouts meant rewinding costs were more than the cost of electricity. This raises the general question of why don't people want a pumpset versus how many people want a pumpset but can't get one.

Do you measure size of wells (borewells), measured by HP (horsepower)? 20 wells isn't comparable across regions. This also links to issues of water-sharing. Local politics and influence can be a factor. This also means larger farmers with pumpsets sell water to their neighbors. Anecdotally, it is prevalent in many eastern states of India.

Is there any analysis of farm size and pumpsets? Many farms in the less irrigated sections of India (easter) are more subsistence, and also (as you rightly observe) have limitations in market access. These regions also have much lower water demands based on water tables. Crop choices also matter – Punjab went for cash crops much earlier on.

An interesting analysis would be to examine districts within a state. One can assume a state has certain policy – but we note there is disparity in pumpsets within states. This suggests it's not a policy reason but fundamentals driven by farmer wealth, crop choices, water tables, rainfall patterns etc. See interior Maharashtra vs. coastal, and similar for some other states.

A fundamental question: What if earlier electrifiers were simply richer regions or had more resources? Thus BOTH homes and pumpsets would be faster. The 8 times more pumpsets for earlier electrifiers then is explained by economic reasons, as opposed to the framing you have, which is a policy choice with a tradeoff.

Figure 3 – we can note that even in 2013, it is the east that lags – both household electrification and irrigation.

Figure 4 – There is a clear split between late vs. early electrifiers, but that split may also have many confounding factors instead of your theory of policies that favored one over the other. Note that there are very few districts with over 5 lakh (500,000) households electrified.

Line 163: "transformer capacities could be constraining even households with the financial means to access electricity for groundwater irrigation." Your math is correct that household connections are small, but transformers are always based on multiple homes, and, more importantly, almost NEVER smaller than tens of kW. Esp. in those days, there were no plans for LVDS (low voltage distribution

systems) which had smaller transformers. Typically, there was a fixed model used that covered a variety of uses. There was no separate transformer then for house vs. agriculture. So size limitations is unlikely to be a bottleneck until we have many pumpsets connected.

In summary: Interest and strong analysis, but the claims made aren't proven, and the econometrics only partially answer some of the issues above (like wealth as a factor - it's not just RURAL HOUSEHOLD WEALTH that matters - state wealth matters for the utility, e.g., the presence of richer consumers to offset rural losses).

Beyond Lights: The Changing Impact of Rural Electrification on Indian Agriculture

Review Responses

Reviewer 1

The central premise of this project is that rural electrification impacts household incomes and a key pathway through which it does so is via enabling households engaged in agriculture to pump groundwater for irrigation. The hypothesis which they proceed to test is that a 2000s turn in policy focus towards household electrification rather than electrification for agriculture, which therefore undermined the growth of electric pumps in agriculture. This “crowding out” of electric agricultural pumps by domestic electrification in turn affected income growth. They proceed to demonstrate this through a series of steps – first, by demonstrating that the relationship of electricity based pumping and domestic access to electricity is on average lower among “late electrifier” districts and the “early electrifier” districts and by showing that rural electrification was not income enhancing for the “late electrifiers” as it was for the “early electrifiers”.

This paper is on a very interesting topic and the authors make very original and creative use of the data. It is motivated very well and the authors produce very useful insights. However, I was left underwhelmed by their bewildering empirical approach and the confusing use of data.

Thank you for your detailed and insightful comments. We have taken your suggestions and added three pieces of analyses to strengthen our claim that electrification timing which captures a change in electrification policy is associated with the slow expansion of agricultural electricity. We understand why in its current form our paper may lead readers to believe that the level of electrification rather than timing may be important in predicting the number of electrified wells. We believe the additional analyses which include - (1) using a different treatment indicator; (2) analysis on a subset of districts with comparable electrification rates and; (3) sub-district analysis within a state, strengthen our claim that it is indeed when electrification took place that impacts where electricity is used for groundwater irrigation. Additionally, to clarify our intent to capture electrification policy change, we have replaced *Late Elec* with *PC* to identify districts that were majorly electrified after the focus of rural electrification policy shifted towards domestic electrification from agriculture. The definition of *PC* is the same as what we previously used for *Late Elec*. We now refer to late electrifying districts as *PC* districts or districts that electrified post policy change, and early electrifiers as non-*PC* districts or districts that electrified pre policy change. We highlight changes in the manuscript by using blue-coloured text. Please see detailed responses to your points below.

The backbone of this paper is the empirical specification in Lines 319-322 that regresses electric pumps (and then acreage irrigated by them) on households with electricity and an interaction term that combines electrified households with and indicator variable denoting if this is greater than 50% of all households. The authors claim the coefficient on the interaction of $elechh_{it}$ with $LateElec_i$, β , is of main interest. It captures the difference θ in the effect of household electrification on pump electrification between early and late electrifying districts. In other words, α captures the relationship between rural electrification and groundwater irrigation among early electrifying districts, while $\alpha + \beta$ captures the estimate for late electrifying districts.

I beg to differ. In essence, this is like a classic structural break model but defined in terms of a 50% threshold based on electrified households. Note that this is exactly the specification one would use to detect threshold

effects that beyond a threshold of electricity diffusion in the village, the relationship between domestic and agricultural use of electricity changes. One could tell several alternate stories with this model: that there is a preference ordering of investment among rural households so that in the early stages of diffusion of electricity people invest in electricity pumps; or a story of village level development – in the early stages of electrification, one invests in agriculture (electric pumps) and eventually as surplus is generated in agriculture, one invests in off farm activities. I simply don't see the empirical specification delivering the story that the authors are trying to get at. Indeed, one could argue just based on the model, that it has little to do with any policy or crowding out, but a story of the evolution or dynamics of electricity use.

We use household electrification to measure rural electrification in our analysis. However, early attempts at rural electrification expansion in India were focused on providing electricity infrastructure to villages with a small number of domestic connections (about 10%) (Banerjee et al., 2014a). It was only in the early 2000s that the focus shifted towards electrifying all households. The target of *Rajeev Gandhi Vidyutikaran Yojana* (RGGVY) were villages with populations greater than a 100 and lacking electricity infrastructure. We find regions electrified under RGGVY to experience limited expansion in agricultural electricity. Therefore it is not how electricity diffuses which impacts its use in agriculture rather when, since villages which got electrified for the first time under RGGVY experienced almost no expansion in agriculture (Banerjee et al., 2014a; Planning Commission of India, 2014; Parikh et al., 2013). We include three specific analyses that strengthen our claim of the relationship between weak expansion of agricultural electricity supply and the timing of electrification -

1. Replace PC_i with $RGGVY_i$ - We replace 50% household electrification threshold captured by PC_i (previously $LateElec_i$ with an indicator variable $RGGVY_i$, which is 1 for districts that were funded under the first wave of electricity expansion under RGGVY during India's X^{th} five year plan spanning 2002-2007. We find our main conclusions to hold with some changes in the precision of the estimates (results included in table 1, columns 5-7). Our reason for not using $RGGVY_i$ in our primary specification is due to issues of data integrity. RGGVY was majorly carried out under the X^{th} (2002-2007), XI^{th} (2007-2012) and XII^{th} (2012-2017) five year plans. However, data for the selection of districts in each wave is unavailable. Even with the limited data, our primary specification picks up the slow expansion of agricultural electricity supply for RGGVY phase I districts. A point to note - the results no longer remain statistically significant with the inclusion of state specific year fixed effects.
2. Sub-district analysis within a state - We also run our primary specification at the block level in Madhya Pradesh. Blocks are administrative units in between villages and districts. Therefore there is variation in the electrification rates within each district, which is another way we test the threshold effect. We chose Madhya Pradesh as it has a high diversity of PC (previously late electrifier) (28) and non-PC (10) (previously early electrifying) districts and is a major agricultural state. We use the Defense Meteorological Program Operation Line Scan System (DMSP-OLS) night time luminosity data compiled by Asher et al. (2021) to measure electrification. We use this additional source as population census does not publish electrification data at the block level or any level lower than districts. We matched the second, third and fifth rounds Minor Irrigation Census with SHRUG nightlight and population census datasets (Asher et al., 2021). The first minor irrigation census does not report data at the block or village level. We use total light luminosity values which range from 0 to 63 and are calibrated by Elvidge et al. (2014) for consistent measure across the range of years between 1994-2013. On average we find DMSP-OLS data to be a conservative estimate of electrification (see figure appendix figure A1). The block-level analysis with DMSP-OLS and Minor Irrigation Census for Madhya Pradesh are consistent with our results overall with some differences in the interpretation due to the use of nightlight luminosity. On average, a 1% increase in luminosity among blocks within non-PC districts (previously early electrifier) is associated with an increase in over 1,500 electrified wells. This value decreases by nearly 1,000 wells and a 1% increase in nighttime luminosity is associated with a cumulative increase of 500 electrified wells in blocks across PC districts (previously late electrifier) in Madhya Pradesh. We do not find a relationship between nighttime luminosity and wells with diesel pumps.

3. District Selection - Finally, to isolate timing rather than proportion of electrification, we select districts from both PC and non-PC districts based on a common window of electrification rates. Figure 1 illustrates our selection process where we choose districts from both groups in the 25% to 50% electrification rates in 1986 for non-PC and in 2001 for PC districts. Using this subset of districts we run our primary specification. For the subset of districts among non-PC districts, we regress the change in electrified wells on change in household electrification between 1986 and 2001. We do the same for PC district subset between 2001 and 2013. Doing so helps us measure similar electrification thresholds between PC and non-PC districts but across different time periods. Results from this analysis have similar implications as our main results (table 1). Across non-PC districts, electrification of 100 households is on average associated with approximately 18 additional wells with electric pumps. Whereas, this number reduces to less than 4 among PC districts.

Figure 1: Selection of districts: (a) All districts; (b) Selected districts.

Table 1: Implications remain the same even for selected districts within common bandwidth of electrification rates

	Groundwater wells		
	Total	With electric pumps	With diesel pumps
Electrified households	0.089** (0.044)	0.180*** (0.051)	0.002 (0.015)
Electrified households \times PC	-0.0832** (0.039)	-0.1408*** (0.049)	0.020 (0.025)
Total households	-0.050 (0.056)	-0.071 (0.053)	-0.015 (0.019)
Fixed-effects			
District	Yes	Yes	Yes
Year	Yes	Yes	Yes
Fit statistics			
Observations	236	236	236
R ²	0.96	0.93	0.93
Within R ²	0.05	0.17	0.03

Note:

*p<0.1; **p<0.05; ***p<0.01

Standard errors are clustered at the district level.

Estimated using equation 1 and a subset of districts from the panel dataset constructed from district-level population and minor irrigation census data.

The authors robustness checks “We also consider models where we include electrification quantiles instead of a single binary classifier and find similar results (see appendix tables A1 and A2), in fact illustrate the diminishing importance of agricultural pump electrification as domestic electrification expands. In short, the authors have an exciting story here (they should write that one), but it is not the one they are trying to tell!

The fundamental problem is “construct” validity. Their bewildering approach to identifying those districts “exposed” to the policy and irrigation oriented versus a domestic oriented electrification policy, named “early” and “late” electrifiers derives not from the time when electrification was introduced but to the extent of electrification (in 2001?) i.e., as “majorly electrified” (lines 116). I struggled to see for myself nor access evidence to see why this is a good construct of early versus late and not more versus less!

We hope that we have been able to convince you as to why we believe that limited expansion of agricultural electricity is related to the timing of electrification and not the extent of electrification with the three additional analyses described above. Additionally, we have also changed $LateElec_i$ to PC_i to better reflect the treatment we measure. PC_i refers to policy change and is defined similarly as $LateElec_i$.

A better approach would be a structural break in terms of time (treat 1986 as a base period and use an indicator variable for post 2000s) to proxy the policy shift. The authors could add a piece-wise analysis based on what proportion of households are electrified.

The issue with using 1986 as a base year and treating all districts similarly is that we are unable to distinguish between districts that majorly electrified before 2001 and those that electrified post 2001 when the target of rural electrification changed from agriculture to domestic connections. Electrified wells have continued to increase in number across both categories of districts till the latest available round in 2013. However, the rate of increase is different and important to isolate, which we are unable to do in the proposed analysis.

I would always want state-year interaction fixed effects to control for time-varying policy changes (including electricity pricing policies, irrigation technology subsidies, other infrastructure, etc.)

We include state-year interaction fixed effects in our primary specification and find similar implications for electrified wells with some changes in the precision of our estimate (tables 2 and 3). Electrification of 100 households among non-PC districts is on average associated with approximately 10 electrified wells compared to nearly 16 in our original analysis. The penalty for electrifying post policy change remains statistically significant and reduces to nearly 3 electrified wells compared to our earlier estimate of 2 additional electrified wells.

Interestingly, although electrification is still associated with a decrease in the use of diesel pumps, the penalty for electrification post policy change no longer remains statistically significant. This is likely due to low cross-state variation in the use of diesel pumps which may be stemming from a loss of statistical power, or perhaps reflective of the dominant use of diesel pumps in alluvial aquifers which are regionally concentrated across India (Shah, 2009). Results for irrigated area also have similar implications for all seasons. On average, electrification of a 100 households is on average associated with increase in nearly 12ha of irrigation during *Kharif* and 10ha during *Rabi* among non-PC districts. These numbers are a bit higher than our previous estimates of 11ha and 9ha respectively. We find post-policy change electrification penalty completely negates any increase in irrigation expansion during *Rabi*, similar to our previous estimates.

Table 2: Household electrification in PC districts is also associated with smaller expansion in irrigated area across all season with variations explained by electrified pumps

	Groundwater wells		
	Total	With electric pumps	With diesel pumps
Electrified households	0.058*** (0.020)	0.101*** (0.025)	-0.028*** (0.008)
Electrified households \times PC	-0.0543** (0.027)	-0.0709** (0.028)	0.0201 (0.013)
Total households	0.0294** (0.014)	0.0036 (0.009)	0.0101 (0.011)
Fixed-effects			
District	Yes	Yes	Yes
State \times Year	Yes	Yes	Yes
Fit statistics			
Observations	969	969	969
R ²	0.74	0.93	0.87
Within R ²	0.12	0.64	0.42

Note:

*p<0.1; **p<0.05; ***p<0.01

Standard errors are clustered at the district level.

Estimated using equation 1 and panel dataset constructed from district-level population and minor irrigation census data.

Table 3: Household electrification in late-electrifiers is also associated with smaller expansion in irrigated area across all season with variations explained by electrified pumps

	Irrigated area (ha)					
	Annual (ha)		Kharif (ha)		Rabi (ha)	
	(1)	(2)	(3)	(4)	(5)	(6)
Electrified households	0.199*** (0.0665)	0.050 (0.0527)	0.117*** (0.0337)	0.037 (0.0253)	0.096*** (0.0320)	0.034 (0.0275)
Electrified Households \times PC	-0.178*** (0.058)	-0.070 (0.043)	-0.063** (0.030)	-0.005 (0.021)	-0.010*** (0.029)	-0.060** (0.025)
Wells with electric pumps		1.492*** (0.182)		0.801*** (0.111)		0.594*** (0.075)
Total households	0.082 (0.060)	0.079 (0.051)	0.026 (0.025)	0.025 (0.021)	-0.005 (0.025)	-0.006 (0.021)
Cumulative Average Monthly Rain†	-11.43 (7.872)	-4.053 (7.458)	-11.91** (4.659)	-6.641 (4.252)	23.00 (35.14)	0.174 (32.52)
Fixed-effects						
District	Yes	Yes	Yes	Yes	Yes	Yes
State \times Year	Yes	Yes	Yes	Yes	Yes	Yes
Fit statistics						
Observations	969	969	969	969	969	969
R ²	0.89	0.90	0.88	0.89	0.89	0.90
Within R ²	0.45	0.51	0.42	0.51	0.42	0.47

Note:

*p<0.1; **p<0.05; ***p<0.01

Standard errors are clustered at the district level.

†Annual, average rainfall during November to March and average rainfall during June to October were used for annual, *Rabi* and *Kharif* cultivation seasons respectively.

Estimated using equation 1 and panel dataset constructed from district-level population and minor irrigation census data.

I would like to see a greater discussion of diesel pumps (lines 66-68) used not as a substitute for electricity but for poor quality of electricity and hours of power supply. There are several discussions of these in the literature on

subsidies in India (notably by Ashok Gulati and co-authors).

Included text in Lines 67-75 - “Electricity is preferred even in shallower wells where the option of pumping using diesel exists and is reflected in the declining proportion of wells operated by diesel pumps from over 30% in 1986 to below 28% in 2013, despite greater than 90% growth in the total number of irrigation wells during the same period (Statistics Division, 2017). At greater than INR 40/l since price deregulation in 2014, diesel is more expensive than agricultural electricity rates in most Indian states.(Verma and Patnaik, 2018; CCEA, 2014)ⁱ Therefore, only in regions where electricity is unavailable or suffers from quality issues of intermittency and low voltage, diesel pumps are used to power irrigation either exclusively, or in conjunction with electric pumps to compensate for unreliable electricity supply (Gulati and Narayanan, 2003; Gulati et al., 2019). Where available, electricity is subsidized by way of flat or no tariffs effectively eroding the marginal cost of consuming electricity (Gulati and Narayanan, 2003).”

Line 23 “Perhaps it is time to reevaluate uniform prescriptive electricity loads and uses in formulating RE policies” This sentence can be clarified. It becomes clear later on but on line 23, it is too early for readers unfamiliar with the policy and the sector.

Changed text in Line 24-27 - “Perhaps it is time to reevaluate domestic electrification as the primary target of RE policies, and instead consider income generation as a primary goal to unleash the true power of electrification in the rural developing world.”

The authors claim that they use “newly constructed panel-dataset spanning three decades” (in the abstract). Yet the figure 2 notes that the “Relationship between electrified households and electrified wells in 2013 was stronger among early electrifiers compared to late electrifying districts. The point estimates represent district-level data on electrified houses and wells with electrified pumps.” This suggests misleadingly that the regression was run for just 1 year, i.e., 2013.

Figure 4 serves as the descriptive evidence to motivate our analysis and is explained by the accompanying text in lines 130-134 - “Before presenting our regression models, figure 4 presents cross-sectional, descriptive evidence to motivate our analysis. In 2013, our most recent round of data, we see a stark difference in the relationship between household electrification and electrified groundwater pumps in PC versus non-PC districts. PC districts had far fewer wells with electric pumps compared to non-PC districts with similar numbers of electrified households (Figure 4). This divergent cross-sectional relationship between the two sets of districts could be driven by unobserved factors.”

We have added the following line (lines 140-141) to clarify that the regression estimates considers three time periods - “Our fixed effects regression estimates measures the association between the number of electrified households with the number of groundwater wells using electric and diesel pumps across 1986 to 2013. The estimates show that, on average, the electrification of ...”

The undertone at least in the initial parts of the paper is a celebration of electricity for pumping groundwater, barring some reflections in line 68. Perhaps you could start the paper early on by suggesting that despite the problem of overextraction etc. irrigation is known to have large benefits, and the culprit for that is more in the pricing than the extraction.

Included text in Lines 9-11 - “This vast consumption of groundwater has not been without its adverse impacts. Every year India pumps twice as much groundwater as the US or China, and houses regions with the greatest rates of global groundwater depletion (Aeschbach-Hertig and Gleeson, 2012; Famiglietti, 2014; Lo et al., 2016).”

ⁱAssuming the energy content of diesel to be 38.9 MJ/l, INR 40/l translates to approximately INR 4/kWh.

Reviewer 2

The authors tackle a very important question with both energy and human development implications.

This is a good dataset and strong analytics – but the interpretations and framings need to be questioned. The study very nicely shows there is a disconnect for irrigation coverage vs. household electrification by time (early vs late), but it's not clear causality can be linked to policies favoring household electrification.

We thank you for your insightful and helpful comments. We have made additions to the manuscript with a new block-level analysis for Madhya Pradesh and text in response to your points and in support of our original claims. Additionally, to clarify our intent to capture electrification policy change, we have replaced *Late Elec* with *PC* to identify districts that were majorly electrified after the focus of rural electrification policy shifted towards domestic electrification from agriculture. The definition of *PC* is the same as what we previously used for *Late Elec*. We now refer to late electrifying districts as *PC* districts or districts that electrified post policy change, and early electrifiers as non-*PC* districts or districts that electrified pre policy change. We highlight changes in the manuscript by using blue-coloured text. Our responses to the individual points are found below.

Utilities likely did slow down connections to farmers, but that was independent of any focus on household electrification. Farmers underpay, as well known, and this non-remunerative supply is a strain on utilities. Note, free power to farmers only started in 1977 in Andhra Pradesh, and wasn't widespread for a few more years.

While electricity access policy DOES favor households, it's not clear why that would be at the expense of agriculture. BOTH segments enjoy heavily cross-subsidized and subsidized electricity (cross-subsides for homes being for lower slabs or tiers of consumption).

It is likely that rural electric supply on average is loss-making in India for the power utilities, whether it is subsidized domestic connections or agricultural supply. It is for precisely this reason, that we think there is some rationing occurring in who gets electricity and where. Together, our results imply that while power utilities or the government may not have deliberately neglected agricultural electricity supply, the singular focus on electricity expansion through domestic connections to fulfill stated targets may have resulted in the unintended neglect of electricity supply to agriculture. We do not have empirically robust methods of teasing out exactly how this may have occurred, but from a synthesis of literature and field visits to Odisha identify two main routes –

1. Transformer sizing – Two independent evaluations of the *Rajiv Gandhi Grameen Vidyutikaran Yojana* (RGGVY) by Parikh et al. (2013) and Planning Commission of India (2014) find frequent burnout of transformers in RGGVY electrified villages. Through surveys, they find overloading (rather than infrastructural defects) as a key reason for these burnouts. Further, Parikh et al. (2013) breakdown the calculation of the standard size of installed transformers of 457kW (16kVA with a power factor of 0.7) load result in average loads of 600W per household (implying average village size of ~ 760 households). Therefore, the average household load is less than the capacity of a 1 HP pump-set (~ 746 W). In such a scenario, nearly 50% of the total calculated load could come from only pumps if even 10% of the houses operated 4HP pump-sets. To place this in context, over 80% of all wells in MI Census 2013 used pumps larger than 4HP (Government of India, 2017)s. Additionally, Banerjee et al. (2014b) also state the limited capacity of transformers as one of the major contributors to low reliability of electricity among RGGVY electrified villages.
2. Distance from transformer and lack of electric infrastructure that provide physical access (poles, wires etc.) for non-domestic uses of electricity - The two studies note voltage drops experienced by far flung houses in villages leading to equipment failure. Additionally, Parikh et al. (2013) find a complete absence of any commercial activities outside of homes in surveyed villages electrified by RGGVY. They find a small number of commercial activities including weaving and handicraft occurring inside homes. Planning Commission of India (2014) find the use of electric motor-pumps for agriculture in less than 1% of households surveyed in RGGVY villages.

We identified similar constraints in field visits conducted during 2016 and 2018 in rural Odisha. Farmers in the

state are expected to pay the additional cost of poles and wiring required to access electricity on their fields (Orissa Lift Irrigation Corporation, 2014; Department of Energy, 2016). Transformer burnouts were also frequently experienced and attributed by the energy department officials to the hooking up of a handful of electric pumps.

One has to start with the reality that agricultural supply is very expensive for the utility – being both subsidized and cross-subsidized. Are there enough takers for a pumpset? Let’s also start with SECC data – 55% of farmers are landless (laborers) so increasing pumpsets helps a subset. Here’s a counterfactual Q: Now that households are 100% electrified, does this now “free up” policy for pumpsets? The pumpsets policy has been the way it is (limited by design, with largesse and political connections important) for a long while in the face of heterogeneity in connectivity.

Data on landless and landed farmers are not available for all years of our analysis and therefore cannot be included in our primary specification. Instead, we conduct a balance test between PC and non-PC districts (previously late and early electrifiers respectively) and find that on average, PC districts are larger with greater numbers of both landed and landless households (table 4). Holding all else equal, greater number of landed households are expected to be associated with greater demand for electrified wells, which we do not find. This makes our results even more stark as we find larger PC districts with fewer electrified wells than the smaller non-PC districts.

Table 4: PC districts on average were larger with more households across all categories.

	Non-PC districts	PC districts	p-value
Average landless households ('000)	178.3621	211.9399	0.04
Average land owning households ('000)	119.9451	165.4355	0.00
Average total land in district ('000 acre)	1,414.8073	2,884.2886	0.02

Data is based on district-level ($N = 425$) Socioeconomic Caste Census carried out in 2011 (Department of Rural Development, 2011).

There is no limited incorporation of the physics, design, and practicality of rural connections. Recent policies (last 10 years) have been towards feeder segregation, but even before this, most states had “rosters” or schedules where agricultural supply was limited to, say, 6 or 8 hours per day. This was controlled through phase-separation, with single phase supply (for homes) meant to be as much as possible, ideally 24 hours in theory (but never in practice).

We are limited in our capacity to carry out an exhaustive analysis of how states operationalize electrification in their respective rural areas. However, in at least one state where rural electrification occurred majorly through RGGVY, farmers are expected to bear the cost of additional poles and wires to use electricity on agricultural fields (Orissa Lift Irrigation Corporation, 2014; Department of Energy, 2016). Further, limited capacities of transformers in RGGVY electrified villages are found to be a major contributor to poor electricity supply (Banerjee et al., 2014b; Planning Commission of India, 2014; Parikh et al., 2013). As noted, the period when feeder separation was largely carried out does not apply to our analysis as the former is being undertaken more recently. Whether separate feeders will be installed in areas where agricultural demand does not currently exist is a question that remains, and one that is important to address.

The second reality that questions the model and framing is how we have “incremental electrification”. Most challenges for household electrification have been with the “last mile connection”. The earlier (and insufficient) definitions of electrification focused on a single lightbulb meant the village was electrified, updated to then be 10% of homes. This was progressively upgraded, which is a good thing, but there is no evidence of a *policy reason* this was at the expense of pumpsets. Your analysis does bring out the point that household supply doesn’t increase wealth much – so it is a separate question of how much one can/should increase pumpsets. If they were charged “full cost” there would be no problem but we know they are not. Money for loss-making utilities is scarce, and so there is a call to be made how much pumpset deployment is appropriate.

Aside: There was also a period (early 2000s) where power supply quality impacted affordability of pumpsets. Even with “cheap power”, frequent burnouts meant rewinding costs were more than the cost of electricity. This raises the general question of why don’t people want a pumpset versus how many people want a pumpset but

can't get one.

Not wanting agricultural pump-sets from the farmers' perspectives can stem from broadly two main reasons. We address the first reason of not needing one by incorporating seasonal rainfall in our analysis. It is also widely accepted that surface water irrigation is insufficient for cultivation during *Rabi* and is increasingly inadequate for cultivation during *Kharif* (Shah, 2009; Shah et al., 2012). The second reason of not wanting may stem from inadequate returns to pump-set investments. We do not address the latter sufficiently in our analysis beyond looking at irrigation access across households of similar wealth across PC and non-PC districts (previously late and early electrifying respectively). We agree that this may be a potential source of error and make a note of it in Lines 227-234. However, if true, our results remain valid as poor electricity quality would then be experienced disproportionately higher by PC (previously late electrifying) districts and would merit further investigations. Not wanting pump-sets from the government's or the utilities' perspectives are important points that are beyond the scope of our study.

Do you measure size of wells (borewells), measured by HP (horsepower)? 20 wells isn't comparable across regions. This also links to issues of water-sharing. Local politics and influence can be a factor. This also means larger farmers with pump-sets sell water to their neighbors. Anecdotally, it is prevalent in many eastern states of India.

Unfortunately, the size of pumps is not provided in the first round of data. We use the classification of wells based on their depths and rerun our primary specification. There are three main types of wells recorded in the Minor Irrigation Census – dug wells which are constructed without the use of drilling machines and have average depths of 8-15m, shallow and deep tube-wells are constructed with drilling machines and are classified as those shallower than and those deeper than 70m respectively. We include these details in the data section under Minor Irrigation Census.

We run the main specification separately for each category of wells and find persistent negative and statistically significant effects among PC districts (table 5). Interestingly, the negative impact of electrifying post policy change is greatest on a relative scale for dug wells, where PC districts experience no gains from electrification in terms of electrified irrigation wells. We are unable to address the issue of water markets due to limitations of data and make a note of this in the discussion section in Lines 231-234 - “We are also unable to account for water markets that are reported to exist in some eastern states in the country (Mukherji et al., 2009). Nevertheless, these factors are important to account for while designing policies that target irrigation expansion across PC districts.”

Is there any analysis of farm size and pumpsets? Many farms in the less irrigated sections of India (easter) are more subsistence, and also (as you rightly observe) have limitations in market access. These regions also have much lower water demands based on water tables. Crop choices also matter – Punjab went for cash crops much earlier on.

As such we are unable to control for the size of farms in our main specification as data on farm size are not available for the years included in our analysis. We do however, control for it when measuring the differences in irrigation across households of similar wealth in PC and non-PC districts. Households with similar consumption in PC districts irrigate less than households in non-PC districts on average, even after controlling for the size of land owned and operated. Additionally, we run a balance test on the surveyed agrarian households included in the Situation Assessment Survey carried out by the NSSO (National Sample Survey Office, 2014). We find that on average, farmers in PC districts do have smaller plots - 1.4 ha compared to 2.2 ha on average in non-PC districts (table 6). On average, PC districts run about half a hectare smaller up until the 75th quartile. So it is possible that a systematic difference in parcel size may be contributing to lower returns on electric pump investment and thereby adding to the disincentives to invest in electric pumps. We have added text in the discussion and policy section to note this in Lines 227-234 - “Our analysis does not exhaustively capture all possible disincentives to electric pump investments - poor supply of electricity, small or scattered parcels of landholdings and inadequate returns to irrigation are some potential biases to our results. For instance, households on average owned smaller parcels of land among PC districts compared to non-PC districts. While we control for land sizes in our analysis,

Table 5: Negative penalty holds for PC districts across all categories of wells

	Groundwater wells with electric pumps		
	Deep tube-wells	Shallow tube-wells	Dug wells
Electrified households	0.045*** (0.008)	0.074*** (0.013)	0.040** (0.018)
Electrified households \times PC	-0.038*** (0.010)	-0.041*** (0.014)	-0.059*** (0.018)
Total households	-0.007 (0.005)	-0.005 (0.004)	0.001 (0.007)
Fixed-effects			
District	Yes	Yes	Yes
Year	Yes	Yes	Yes
Fit statistics			
Observations	969	969	969
R ²	0.52	0.78	0.87
Within R ²	0.14	0.18	0.04

Note:

*p<0.1; **p<0.05; ***p<0.01

Standard errors are clustered at the district level.

Estimated using equation 1 and panel dataset constructed from district-level population and minor irrigation census data.

we are unable to consider poor quality of supply, scattered parcels or returns to irrigation due to data limitations. We are also unable to account for water markets that are reported to exist in some eastern states in the country (Mukherji et al., 2009). Nevertheless, these factors are important to account for while designing policies that target irrigation expansion among PC districts.”

Table 6: Farm sizes were smaller on average in PC districts in 2013

	Non-PC districts	PC districts	p-value
Land owned (ha)	2.23	1.42	0.00
Land operated (owned and leased) (ha)	2.61	1.55	0.00
Land cultivated during Rabi	1.66	1.01	0.00
Land cultivated during Kharif	2.09	1.17	0.00

Data is based on a nationally representative survey data of agricultural households ($N = 11,182$) in 2012-13 (National Sample Survey Office, 2014).

An interesting analysis would be to examine districts within a state. One can assume a state has certain policy – but we note there is disparity in pumpsets within states. This suggests it’s not a policy reason but fundamentals driven by farmer wealth, crop choices, water tables, rainfall patterns etc. See interior Maharashtra vs. coastal, and similar for some other states.

We run our primary specification at the block level in Madhya Pradesh. We chose Madhya Pradesh as it has a high diversity of PC (28) and non-PC (10) districts and is a major agricultural state. We use the Defense Meteorological Program Operation Line Scan System (DMSP-OLS) night time luminosity data compiled by Asher et al. (2021) to measure electrification. We use this additional source as population census does not publish electrification data at the block level. We matched the second, third and fifth rounds Minor Irrigation Census with SHRUG nightlight and population census datasets (Asher et al., 2021). The first minor irrigation census does not report data at the block or village level. We use total light luminosity values which range from 0 to 63 and are calibrated by Elvidge et al. (2014) for consistent measure across the range of years 1994-2013.

On average we find DMSP-OLS data to be a conservative estimate of electrification (see appendix figure A1). The block-level analysis with DMSP-OLS and Minor Irrigation Census for Madhya Pradesh are consistent with our results overall with some differences in the interpretation due to the use of nightlight luminosity. On average, a 1% increase in luminosity among blocks within non-PC districts is associated with an increase in over 1,500

electrified wells. This value decreases by nearly 1,000 wells and a 1% increase in nighttime luminosity is associated with a cumulative increase of 500 electrified wells in blocks across PC districts. We do not find a relationship between nighttime luminosity and wells with diesel pumps.

Additionally our analysis using the National Sample Survey Office’s Situation Assessment Survey of Agricultural Households data in section titled **“Income does not explain irrigation differences between PC and non-PC districts”** considers the roles that farmer wealth and rainfall play and we find that rural electrification policy change is a stronger predictor of irrigation area than is farmer wealth. Compared to the poorest 15th percentile households, all other farming households among non-PC districts irrigated larger tracts of cultivated land during Rabi. We find no similar differences in irrigated area between the poorest 15th percentile households in non-PC districts and households among PC districts that were even in the top 85th consumption percentile. The NSSO analysis includes state-level fixed effects which translates to a comparison of agrarian households across PC and non-PC districts within each state. However, the state level fixed effects work only when states with more PC districts do not systematically differ from states with fewer PC districts.

A fundamental question: What if earlier electrifiers were simply richer regions or had more resources? Thus BOTH homes and pumpsets would be faster. The 8 times more pumpsets for earlier electrifiers then is explained by economic reasons, as opposed to the framing you have, which is a policy choice with a tradeoff.

We agree that systematic differences between PC and non-PC districts are potential sources of biases in our results. We conduct a household level analysis (results illustrated in figure 5) where we test for irrigation coverage among households of similar wealth but compare across PC and non-PC districts. We find that the differences persist even among households of similar wealth across PC and non-PC districts. Unfortunately we cannot test this across time since situation assessment of agricultural households has not been repeated since 2013. As you rightly point out in your previous comment, it could be that district-level wealth may be a stronger predictor of pump-set use than household wealth, which this analysis would not address. Our new block-level analysis in Madhya Pradesh compares blocks across PC and non-PC districts. While it does not fully address a possibility of systematic differences across the two categories of districts, it does remove state-level differences that may be present in our primary specification.

If, in fact, non-PC districts are systematically different from PC districts in ways that influence both electrification of houses and wells, while the precision of our estimate may be questionable, our analysis is helpful in at least identifying that irrigation expansion remains unaffected despite rural electrification expansion in the latter. A point that is important to be addressed to tackle rural poverty and climate change.

Figure 3 – we can note that even in 2013, it is the east that lags – both household electrification and irrigation.

We agree it is the east where both rural electrification has been slow to spread and so has groundwater irrigation. Recent estimates suggest near universal electrification of this region ((see *Saubhagya* dashboard). It will be interesting to note the associated expansion in groundwater irrigation once the latest round of Minor Irrigation Census becomes publicly available.

Figure 4 – There is a clear split between late vs. early electrifiers, but that split may also have many confounding factors instead of your theory of policies that favored one over the other. Note that there are very few districts with over 5 lakh (500,000) households electrified.

We agree there could be a number of confounding factors which is why our estimates are based on a two-way fixed effects regression to control for the confounders and explain the details of the estimation strategy in methods section of the paper. Figure 4 simply serves as the descriptive evidence to motivate our analysis and is explained by the accompanying text in lines 130-134 - “Before presenting our regression models, figure 4 presents cross-sectional, descriptive evidence to motivate our analysis. In 2013, our most recent round of data, we see a stark difference in the relationship between household electrification and electrified groundwater pumps in PC versus non-PC districts. PC districts had far fewer wells with electric pumps compared to non-PC districts with similar numbers of electrified households (Figure 4). This divergent cross-sectional relationship between the two sets of districts could be driven by unobserved factors...”

We have added the following line (lines 140-141) to clarify that the regression estimates considers three time

periods - “Our fixed effects regression estimates measures the association between the number of electrified households with the number of groundwater wells using electric and diesel pumps across 1986 to 2013. The estimates show that, on average, the electrification of ...”

Line 163: “transformer capacities could be constraining even households with the financial means to access electricity for groundwater irrigation.” Your math is correct that household connections are small, but transformers are always based on multiple homes, and, more importantly, almost NEVER smaller than tens of kW. Esp. in those days, there were no plans for LVDS (low voltage distribution systems) which had smaller transformers. Typically, there was a fixed model used that covered a variety of uses. There was no separate transformer then for house vs. agriculture. So size limitations is unlikely to be a bottleneck until we have many pumpsets connected.

Calculations by Parikh et al. (2013) suggest the standard size of installed transformers of 457kW (16kVA with a power factor of 0.7) load result in average loads of 600W per household (implying average village size of ~ 760 households). Therefore, the average household load is less than the capacity of a 1 HP pump-set ($\sim 746W$). This would mean that even if 10% households were to own a 4HP pumpset, they would account for nearly 50% of the total village load. Over 80% of all wells in MI Census 2013 used pumps larger than 4HP (Government of India, 2017). Additionally Banerjee et al. (2014b) report on Indian electrification experience notes that many states installed transformers that fall short of full village load leading to issues of reliability.

In summary: Interest and strong analysis, but the claims made aren't proven, and the econometrics only partially answer some of the issues above (like wealth as a factor - it's not just RURAL HOUSEHOLD WEALTH that matters - state wealth matters for the utility, e.g., the presence of richer consumers to offset rural losses.

We added state specific time trends to account for time variant state-level differences in our primary specification. Our results are robust to the inclusion of these state specific time trends (tables 2 and 3). Additionally, the new block level analysis carried out for Madhya Pradesh implies similar differences in block level outcomes between PC and non-PC districts.

We hope that we have convinced you with the new set of analyses which, in our view, add robustness to our claim that differences in the number of electrified wells is independent of household and state wealth, and the need for irrigation.

References

- Aeschbach-Hertig, W. and Gleeson, T. (2012). Regional strategies for the accelerating global problem of groundwater depletion. Nature Geoscience, 5(12):853–861.
- Asher, S., Lunt, T., Matsuura, R., and Novosad, P. (2021). Development Research at High Geographic Resolution: An Analysis of Night Lights, Firms, and Poverty in India using the SHRUG Open Data Platform. The World Bank Economic Review.
- Banerjee, S. G., Barnes, D. F., Mayer, K., Samad, H. A., and Singh, B. N. (2014a). Chapter 2: Closing the Electricity Access Gap. In Power for all : electricity access challenge in India. World Bank Group, Washington, DC.
- Banerjee, S. G., Barnes, D. F., Mayer, K., Samad, H. A., and Singh, B. N. (2014b). Chapter 4: History of Rural Electrification and Institutional Organization. In Power for all : electricity access challenge in India. World Bank Group, Washington, DC.
- CCEA, C. C. o. E. A. (2014). Deregulation of Diesel Prices. Press Release, Press Information Bureau (PIB), Government of India, New Delhi.
- Department of Energy, G. o. O. (2016). Office Memorandum - Amendment of Biju Gram Jyoti-Rural Electrification programme of the State Government.
- Department of Rural Development, Ministry of Rural Development, G. o. I. (2011). Socio Economic and Caste Census 2011 (SECC).
- Elvidge, C. D., Feng-Chi, H., Baugh, K. E., and Ghosh, T. (2014). National trends on satellite-observed lighting. Global urban monitoring and assessment through earth observation, (23).
- Famiglietti, J. S. (2014). The global groundwater crisis. Nature Climate Change, 4(11):945–948.
- Government of India, Ministry of Water Resources, R. D. a. G. R. (2017). 5th Census of Minor Irrigation Schemes Report.
- Gulati, A. and Narayanan, S. (2003). Power Subsidies. In The Subsidy Syndrome in Indian Agriculture. Oxford University Press, New Delhi.
- Gulati, A., Sharma, B., Banerjee, P., and Mohan, G. (2019). Getting More from Less: Story of India’s Shrinking Water Resources. Technical report, Indian Council for Research on International Economic Relations (ICRIER), Delhi.
- Lo, M.-H., Famiglietti, J. S., Reager, J. T., Rodell, M., Swenson, S., and Wu, W.-Y. (2016). GRACE-Based Estimates of Global Groundwater Depletion. In Tang, Q. and Oki, T., editors, Terrestrial water cycle and climate change: natural and human-induced impacts, number 221 in Geophysical monograph series. AGU, American Geophysical Union, Washington, D.C.
- Mukherji, A., Das, B., Majumdar, N., Nayak, N., Sethi, R., and Sharma, B. (2009). Metering of agricultural power supply in West Bengal, India: Who gains and who loses? Energy Policy, 37(12):5530–5539.
- National Sample Survey Office, M. o. S. a. P. I. (2014). Situation Assessment Survey of Agricultural Households - 2013. Technical report, Government of India, New Delhi.
- Orissa Lift Irrigation Corporation, G. o. O. (2014). Detail Information about Jalnidhi-I.
- Parikh, J., Dutta Biswas, C., and Panda, R. R. (2013). Combined Report on Evaluation of Rajiv Gandhi Grameen Vidyutikaran Yojana (RGGVY) of Rajasthan, Assam, Gujarat, Himachal Pradesh and Uttar Pradesh. Technical report, Integrated Research and Action for Development (IRADe), New Delhi.
- Planning Commission of India (2014). Evaluation Report on Rajiv Gandhi Grameen Vidyutikaran Yojana (RGGVY). 224, Programme Evaluation Organisation, Planning Commission, Government of India, New Delhi.

- Shah, T. (2009). Climate change and groundwater: India's opportunities for mitigation and adaptation. Environmental Research Letters, 4(3):035005.
- Shah, T., Giordano, M., and Mukherji, A. (2012). Political economy of the energy-groundwater nexus in India: exploring issues and assessing policy options. Hydrogeology Journal, 20(5):995–1006.
- Statistics Division, M. I. (2017). Manual For Data Collection in the Census of Minor Irrigation Schemes 2013-14. Technical report, Ministry of Water Resources, Government of India, New Delhi.
- Verma, N. and Patnaik, S. (2018). Graphic: India's petrol, diesel prices surge to record. Reuters.

Reviewers' Comments:

Reviewer #1:

Remarks to the Author:

Thank you for the efforts you have made in addressing each comment from the reviewers seriously and as comprehensively as your dataset would allow.